# READING THE ROOM: LEARNING GROUP STATES BEYOND POOLED INDIVIDUAL SIGNALS

## ABSTRACT

The fundamental challenge in modeling group dynamics is that collective states arise from interdependent processes that violate the standard assumption of independent observations. We study this in a triadic Collaborative Problem-Solving (CPS) task where five constructs (group synchrony, group confidence, group interaction phase, individual engagement, individual leadership) are annotated as directional trends (increase / stable / decrease) over short windows. We formulate a hypothesis that group-level states are not reliably recovered from pooled individual features due to the aggregation fallacy. To test this, we introduce Syntality, a benchmark with participant-indexed multimodal streams and paired individual+group trend labels, and SyntalNet, an architecture that satisfies three minimal requirements: (i) permutation-equivariant cross-participant fusion, (ii) mask-aware intra-modality fusion, and (iii) low-rank cross-modal interactions. On Syntality, SyntalNet consistently outperforms additive baselines, improving group-level macro-F1 from 0.37–0.49 to 0.62 while also achieving strong individual-level performance (e.g., 0.64 balanced accuracy / 0.85 AUROC / 0.63 F1 on engagement; 0.60 / 0.78 / 0.58 on leadership). Under 5-fold cross-validation we show that gains are statistically significant. In a leave-one-group-out (LOGO) setting, zero-shot macro-F1 on held-out teams recovers to 0.47 when fine-tuning on 25% of the target group, and to 0.58 when freezing the encoder and adapting only classifier heads. Ablation studies confirm that explicit cross-participant fusion, intra-modality fusion, and low-rank cross-modal fusion each contribute to robustness under various corruption scenarios. Critically, our results provide empirical evidence that pooled individual signals yield performance statistically equivalent to constant predictors on group states in this triadic CPS setting, highlighting the necessity of explicit cross-participant modeling. We will release our dataset with processed features, code repository, and models upon acceptance.

## 1 INTRODUCTION

In group dynamics, observations are interdependent: each participant's state depends on others'. Most multimodal systems still model individuals in isolation or pool features, discarding cross-participant dependencies (Wohltjen & Wheatley, 2024; Tomashin et al., 2022; Kim et al., 2024; Ouyang et al., 2023). While collective intelligence research has established the non-additivity of group-level constructs (Woolley et al., 2010; An et al., 2025), this mismatch between problem structure and modeling assumptions leads to systematic underperformance on group-level prediction tasks.

We identify four classes of existing approaches and their fundamental limitations: (i) *Aggregation methods* that compute group features as $f(\sum_i \mathbf{x}_i)$ or $f(\frac{1}{n}\sum_i \mathbf{x}_i)$, which are subject to the aggregation fallacy and yield near-constant performance on group-level targets; (ii) *Synchrony-based features* that capture pairwise correlations in turn-taking, gaze, or physiological signals, but lack robustness across interaction contexts; (iii) *Sequential and attention models* that process participant streams independently before fusion, often omitting permutation invariance or modality missingness handling; and (iv) *Language model approaches* that process only textual transcripts, discarding prosodic, temporal, and visual cues essential for group state inference (Zhang et al., 2023; Mohamed & Lemaignan, 2021). Empirically, pooled-signal OLS is near-constant on group targets and correlations are <0.18 (Fig. 4), motivating explicit cross-participant modeling. These limitations motivate

three key architectural requirements: permutation-equivariant cross-participant fusion, principled handling of missing modalities, and explicit modeling of inter-participant dependencies.

We formalize an under-tested conjecture: in small co-located CPS teams, additive composition of independently encoded participants is not sufficient for accurate group-state prediction. Concretely, we compare the "additive family" of models that encode each participant $p$ independently ($\phi(x_p)$) and pool via $g\left(\frac{1}{n}\sum_p \phi(x_p)\right)$ against architectures with explicit cross-participant fusion prior to pooling. Our analysis in Section 6 shows that, on Syntality, additive models behave similarly to constant predictors on group-level targets, whereas cross-participant fusion yields large and statistically significant gains.

We propose SyntalNet, which addresses these requirements through three principled components: (i) mask-aware encoders that handle missing modalities via learned masking strategies; (ii) Social Squeeze-and-Excitation (SoSE-X) modules that implement permutation-equivariant cross-participant context gating; and (iii) low-rank cross-modal fusion (GLRX) that captures inter-modal dependencies while maintaining computational efficiency. The architecture culminates in DeepSets-compatible prediction heads that preserve permutation invariance. Empirically, this design yields statistically significant improvements over temporal CNN, LSTM, and vision-language baselines, while our analysis confirms that pooled individual features perform equivalently to constant predictors on group-level targets.

We evaluate on Syntality (10 triads; 170 min) and predict directional trends over consecutive clip-pairs (10-s windows, 3-s stride; 19-s observation).

**Our contributions are:**

- **Formulation & Dataset.** We formulate directional group-state trends (increase / stable / decrease) for five constructs at both individual and group levels, and release Syntality, a triadic CPS benchmark with participant-indexed multimodal streams, temporally smoothed trend labels, and inter-rater agreement statistics.
- **Hypothesis and Architecture**: We state and test the hypothesis that additive pooling of individual features is insufficient for group-state prediction in this setting, and derive minimal architectural requirements. SyntalNet instantiates these requirements via SoSE-X (cross-participant fusion), BSC-X (mask-aware intra-modality fusion), and GLR-X (low-rank cross-modal fusion) with DeepSets heads.
- **Empirical Evidence**: Through K-fold CV, comprehensive ablations, and LOGO experiments, we show that: (i) additive baselines underperform models with explicit cross-participant fusion, (ii) BSC-X and GLR-X improve robustness to data corruption scenarios, and (iii) SyntalNet's encoder supports few-shot adaptation to unseen teams with distribution shifts (OOD).

On Syntality, SyntalNet achieves 0.64/0.60 balanced accuracy and 0.85/0.78 AUROC on individual engagement/leadership respectively, outperforming the Temporal CNN baseline.

## 2 RELATED WORK

### 2.1 GROUP COLLABORATION

Group performance cannot be reduced to the sum of individual abilities Laughlin et al. (2002). Collective states such as cohesion, synchrony, and shared intent emerge from dynamic interpersonal processes rather than individual attributes Salas et al. (2008); Marks et al. (2001). Teams with similar individual skills may differ widely in outcomes depending on coordination, communication, and mutual responsiveness Cooke et al. (2013).

Research has examined individual states such as trust, cognitive load, and attention Healey & Picard (2005); Lee & See (2004), but these alone do not capture the collective constructs that determine group effectiveness. Pioneer work in social signal processing highlights the role of nonverbal behavior in shaping group dynamics, e.g. dominance and leadership inferred from speaking and gaze patterns Jayagopi et al. (2009); Sanchez-Cortes et al. (2010; 2013). Group states, such as cohesion, engagement, synchrony, and leadership emergence, are increasingly recognized as key predictors of team performance Gordon (2025); Hung & Chittaranjan (2010); Ohayon & Gordon (2024). Cohesion supports persistence in problem solving Beal et al. (2003), smooth turn-taking improves efficiency Haan et al. (2021), and leadership emerges through shifting roles Chi & Wylie (2014).

## 2.2 Multimodal Sensing of Human and Group States

Broader surveys emphasize that collaboration relies on multimodal coordination of verbal, visual, and physiological cues Vinciarelli et al. (2008); Beyan et al. (2023). Multimodal sensing provides powerful tools for quantifying internal states. Audio-visual and physiological signals capture stress, trust, affect, and cognitive load (Niu et al., 2024). Speech prosody and facial expressions reveal emotion and engagement (Xie et al., 2025), gaze and gestures reflect coordination (Qi et al., 2025), and physiological measures such as heart rate and skin conductance indicate arousal and workload Zhou et al. (2025).

Recent work extends multimodal sensing to group states in team interactions. Early studies aggregated individual features (e.g., mean arousal in a team) as group proxies (Praharaj et al., 2021), but these offered limited predictive power by ignoring relational dynamics. More advanced approaches explore synchrony measures, such as speech alignment, gaze, or physiological rhythms (Tomprou et al., 2021), as well as collective features such as group-level turn-taking statistics (Hu & Chen, 2022). While promising, these methods still underrepresent the bidirectional influences between individuals and the group, motivating frameworks that explicitly model cross-agent interactions. For instance, attention mechanisms and relational graph models have been used to infer leadership and engagement patterns from multimodal data (Maman et al., 2020; 2021).

Recent work in probabilistic and multilevel dynamic modeling also examines interpersonal coordination in multi-party settings ((Soares et al., 2024; Moulder et al., 2022), using multilevel vector autoregression and probabilistic graphical models to capture within- and between-person dynamics. These approaches provide interpretable accounts of temporal dependencies, but typically operate on low-dimensional, hand-engineered features rather than high-dimensional multimodal streams.

In parallel, large language models (LLMs) have opened new opportunities for sensing group interaction. They have been applied to analyze discussion transcripts (Tran et al., 2024), extract markers of leadership and engagement (Saeid et al., 2024), and even predict group outcomes from linguistic patterns (Kartigueyan & Salisbury, 2025). Yet, existing work remain text-only, overlooking nonverbal cues such as prosody and synchrony that are central to collaboration.

Prior work provides tools for individual sensing, but few frameworks jointly encode multimodal inputs across teammates while preserving their relational structure. Our work builds directly on this gap, proposing a sensing architecture designed to model cross-agent dependencies and to capture the emergent constructs of group collaboration.

## 3 Dataset and Annotation

There are many public datasets for human behavior understanding, in scenarios like human collaborative problem solving and cognitive tasks, such as group brainstorming, and board games Beyan et al. (2023). Collaborative cognitive tasks with a physical aspect, such as joint assembly or object manipulation, provides a rich substrate for understanding coordination, mutual adaption, and nonverbal communication among humans. These tasks inherently require tightly coupled temporal and spatial alignment of actions across agents, making them particularly suitable for studying multi-agent behavioral dynamics.

Datasets such as ELEA, GAME-ON, UDIVA, and Idiap Wolf provide rich multimodal recordings of dyadic or group interactions, often with global group labels (e.g., performance, cohesion) or sparse event codes. However, they typically lack (i) triadic co-located CPS tasks with participant-indexed multimodal streams and (ii) paired individual+group directional trend labels for multiple constructs. Thus, Syntality extends the Weights Task Dataset (Khebour et al., 2024) specifically to fill this gap and to enable the hypothesis test in Section 1. The original dataset comprises triads completing block weight guessing tasks at a round table. We further annotate individual and group construct labels through a crowd-sourcing study and process the multimodal data streams. There are a total of 10 triads that completed the experiment. More details about the task can be found in A.2.

### 3.1 Signal Processing & Data Preparation

The original dataset primarily comprised multi-view audiovisual recordings, body joint positions/orientations, and utterance transcripts to capture verbal exchanges and gross nonverbal behaviors. In Syntality, we add new annotated streams for richer multimodal prediction power. We release

uniform, model-ready representations while preserving the nuanced phenomena and avoiding pre-processing bias. Full pipeline details are in A.4.

*Videokinetics*. We use three synchronized camera views (one per participant). Individuals are tracked and cropped into person-centered streams, temporally aligned and normalized for resolution and frame rate (Appendix A.4.1). Kinect skeletons are synchronized and mapped to participants via IDs, expressed in a body-centered frame; given seated participants, we retain only upper-body joints, interpolate short gaps, and smooth trajectories (Appendix A.4.2). When a primary view is missing, we borrow an alternate view and apply rigid Kabsch alignment on upper-body anchors before re-centering. Facial signals (OpenFace landmarks, head pose, gaze) are linked to participants, normalized for translation/scale, and temoprally stabilized (Appendix A.4.3).

*Dialogue*. This modality comprises utterances and turn-taking streams. ASR transcripts (Google Cloud + Whisper) are manually corrected, diarized per participant, and time-stamped. From them we derive group-level turn dynamics (positions, durations, pauses, speaker changes, overlaps). Overlaps are labeled as *floor-taking*, *butting-in*, or *backchannels*; backchannels are adjudicated automatically by an LLM that inspects the overlapping fragment in the context of the host utterance. Dialogue streams are time-aligned with all other modalities (Appendix A.4.4, A.4.5).

*Acoustic*. We extract speech features with Parselmouth Jadoul et al. (2024) (pitch, HNR, MFCCs, energy, jitter, shimmer, silence ratio) and sentiment posteriors from SUPERB Yang et al. (2021) (neutral / happy / sad / angry). These cues capture stress and arousal (jitter, low HNR), engagement/dominance (energy), and coordination breakdowns (silence) Voleti et al. (2019); Meza et al. (2023). All acoustic features are computed on $10\,\text{s}$ windows aligned with the video clips (Appendix A.4.6).

## 3.2 CROWD-SOURCING ANNOTATION

To have individual and group state labels in the Syntality dataset, we designed an online study. Each participant observed 10 video clips; each clip is $10\,\text{s}$ long and is drawn from a sliding window over the session with a $3\,\text{s}$ stride, following thin-slice work that uses segments of such lengths for reliable judgments of group constructs (Ambady & Rosenthal, 1992). Each clip was annotated by at least 3 participants, who answered 5 questions after each clip, as shown in A.3. We selected these states and the according questions from the current literature, including group cohesion Hoegl & Gemuenden (2001), confidence Guzzo et al. (1993), interaction phase Miller (2003), engagement Schaufeli et al. (2003), and leadership role Aarons et al. (2014). We recruited a total of 1128 annotators, on the online research platform Prolific.

We evaluated inter-rater agreement using percent agreement and considered a label strong if agreement for that sample exceeded 66.7%. The strong label rates for the 5 classes were 80.37%, 79.20%, 76.71%, 91.44%, 89.57%. The Fleiss' $\kappa$ for the 5 classes were 0.286, 0.284, 0.348, 0.439, and 0.468, demonstrating fair to moderate agreement. To address weak labels, particularly for group states, we applied kernel label smoothing. Discrete annotation with agreement scores were treated as soft distributions, then convolved with a confidence weighted Gaussian kernel to yield temporally coherent posteriors that reduce noise and handle gaps.

Since state changes are often more actionable than raw levels Xu et al. (2021), we derive labels as trends (increase/stable/decrease) between pairs of clips. Clips are $10\,\text{s}$ long and are extracted on a $3\,\text{s}$ grid. Each trend label reflects the change in the temporally smoothed posteriors over an effective $19\,\text{s}$ observation window with $1\,\text{s}$ overlap between the two clips. Distributions of the resulting trend labels are depicted in Figure 8.

Statistical analysis of the Syntality dataset showed weak correlation between individual and group states, but strong links between states and synchronized behaviors (e.g., mutual attention, shared gaze). Visualization and details are in A.5, supporting our hypothesis that group states are best modeled through participant interplays.

## 4 PROBLEM SETUP

We aim to model the *evolution of group dynamics* during a short interaction as an emergent state by predicting the *directional change* (increase, stable, or decrease) for five dynamic constructs observed within a *clip C* (duration = 19s). Two are *individual-level* constructs and three are *group-level*. The trend labels are derived from annotations described in section 3.2 and are treated as ground

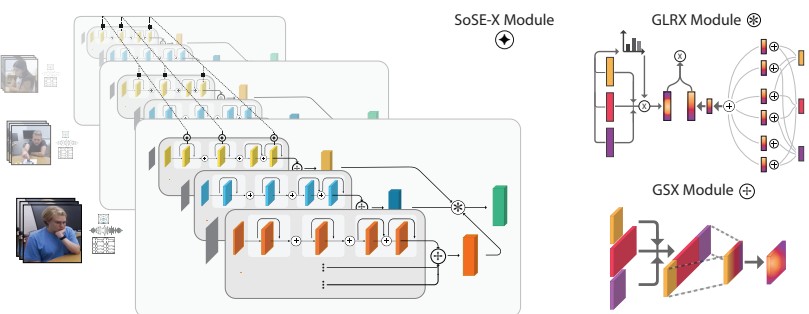

Figure 1: Syntal architecture: Each participant's multimodal inputs are first encoded into masked feature sequences. Fusion occurs through three mechanisms: BSC-X (intra-modality fusion) integrates heterogeneous substreams within each modality; SoSE-X (cross-participant fusion) exchanges salient features across individuals in a permutation-equivariant way, embedding interpersonal context; and GLR-X (cross-modal fusion) combines an mask-aware gate with low-rank pairwise interactions, ensuring robust integration across modalities even when some are absent.

truth. Let $P = \{1, \ldots, |P|\}$ be the participants and $\mathcal{M}$ the set of sensed modalities (e.g., videokinetic, dialogue, acoustic). For person $p$ and modality $m \in \mathcal{M}$, the time-aligned feature stream and modality mask over $C$ is $\mathbf{X}_p^{(m)} \in \mathbb{R}^{1 \times T_m \times k_m}$ and $\mathbf{M}_p^{(m)} \in \mathbb{R}^{1 \times T_m \times k_m}$ respectively, where $k_m$ is the feature dimensionality of modality $m$ and $T_m$ is the temporal or event-based sequence for that modality. The model takes as input the multimodal behavioral streams of all participants in a clip $C$, $\mathbf{X} = \{\mathbf{X}_p^{(m)} : p \in P, m \in \mathcal{M}\}$. Because $P$ is an unordered set, the function must be permutation-invariant with respect to participants. The objective is to learn a mapping $F : \mathbf{X} \mapsto (\mathbf{y}_I, \mathbf{y}_G)$, where $\mathbf{y}_I = \{y_{i_1,p}, y_{i_2,p}\}_{p \in P}$ are the individual-level cohesion trends and $\mathbf{y}_G = \{y_{g_1}, y_{g_2}, y_{g_3}\} \in \{-1, 0, +1\}^3$ are the group-level trends. Each label $y \in \{-1, 0, +1\}$ denotes a *decrease*, *stable state*, or *increase* in cohesion for the corresponding construct. All predictions (group- and individual-level) are made solely from person-level behavioral streams, enabling proactive insights into how group-affective emergent states evolve during short interactions.

# 5 MODEL: SYNTALNET

We present SyntalNet (SYNTALity NETwork), a multimodal architecture that captures the temporal evolution of behavioral cues in small-group interactions by fusing heterogeneous, time-synchronized signals while preserving interpersonal structure (Figure 1). SyntalNet infers changes in dynamic constructs within each clip of group interactive sessions by learning individual-level representations. It combines ConvNeXtV2-style backbones (Woo et al., 2023) with partial convolutions (Liu et al., 2018) to encode missing/varying-length streams, and employs specialized fusion mechanisms to model interpersonal, within-modality, and cross-modality dependencies.

Each modality adopts a multi-pathway design for various data streams, including *low-dimensional feature* streams that aggregate structured cues, and *high-dimensional embedding* streams that carry richer semantics. This separation prevents the compact, highly informative signals from being overwhelmed by the higher-capacity descriptors, while giving the fusion layers room to adaptively rebalance contributions downstream in proportion to their sample-wise utility.

## 5.1 MODALITY-SPECIFIC INPUTS

In the **Videokinetics** branch, the low-dimensional pathways are facial and skeletal measurements, including $\mathbf{X}^{\text{face}} \in \mathbb{R}^{212}$ (gaze angles, 6-DoF head pose, and $68 \times 3$ 3D landmarks) and $\mathbf{X}^{\text{pose}} \in \mathbb{R}^{182}$ (26 joints with 3D positions and quaternion orientations). The high-dimensional pathway processes $\mathbf{X}^{\text{video}} \in \mathbb{R}^{1024}$, extracted per clip from the avg_pool3d intermediate endpoint of an I3D (Inception-V1) encoder pretrained on ImageNet; as the video representation (see Appendix A.4.1-A.4.3).

In the **Dialogue** branch, low-dimensional turn-taking dynamics are summarized as $\mathbf{X}^{\text{turns}} \in \mathbb{R}^{12}$, capturing conversational flow (e.g., talk/silence spans, overlaps, etc.), see Appendix A.4.4. High-level utterance semantics are processed as $\mathbf{X}^{\text{utterance}} \in \mathbb{R}^{1024}$, extracted by last-token pooling (EOS) of Llama 3.2 Large Language Model; as the sentence-level embedding (see Appendix A.4.5).

In the **Acoustic** branch, prosodic and affective cues are the stacks of the low-dimensional stream as $\mathbf{X}^{\text{prosody}} \in \mathbb{R}^7$ (pitch, HNR, MFCC, energy, jitter) and $\mathbf{X}^{\text{sentiments}} \in \mathbb{R}^5$ (angry, happy, neutral, sad, silence). The high-dimensional stream $\mathbf{X}^{\text{audio}} \in \mathbb{R}^{512}$ is obtained from a pretrained `pyannote.audio` encoder, providing a summary of short-term timbre and phonetic content (see Appendix A.4.6).

All streams are time-aligned per participant and accompanied by validity masks. Subsequent layers operate with *partial* convolutions and pooling, which apply only to valid locations and propagate masks across layers. Furthermore, since feature dimension $F$ is an unordered set of heterogeneous channels, we therefore avoid locality-biased operations by using kernels with unit size over this dimension (unless explicitly stated).

## 5.2 ENCODER BACKBONE

The per-pathway encoder stacks $S = 3$ stages following a partial $1 \times 1$ convolution in the stem to project the channels to that of first stage $C_1$. Each stage is instantiated by shallow ConvNeXtV2-style blocks with depth $D_s \in \{1, 2\}$ performing channels-wise layer normalization, a partial depthwise convolution, a pairwise expansion (bottleneck), regularization, nonlinear activation, and residual connections. In our main configuration, we do not pool over $T$ until downstream fusion so as to preserve frame-level resolution and capture fine-grained temporal dynamics.

## 5.3 CROSS-PERSON FUSION

In order to capture social and behavioral interdependencies among participants, we introduce *Social Squeeze-and-Excitation–based Cross Fusion (SoSE-X)*. Inspired by Squeeze-and-Excitation (SE), we generalize channel re-weighting from a single stream to multi-person settings by coupling *within-person* saliency with *cross-person* context injection. For each participant's feature map $Z_p \in \mathbb{R}^{C \times T}$, SoSE-X first applies a concurrent channel–temporal SE gate: global average pooling over $T$ followed by a bottleneck MLP for channel scores and a $1 \times 1$ convolution for a temporal score; the two are summed and passed through a sigmoid to yield a dense mask $G_p \in (0, 1)^{C \times T}$. Thus, we distill each person's most informative channels/locations, reducing noise propagation across persons. The gated features $\tilde{Z}_p = Z_p \odot G_p$ are then fused across people via the mean of others, and added back to the original stream with a learnable scalar residual $\alpha$:

$$\bar{Z}_p^{\text{ctx}} = \frac{1}{|\mathcal{N}(p)|} \sum_{q \in \mathcal{N}(p)} \tilde{Z}_q, \quad \mathcal{N}(p) := P \setminus \{p\}, \quad Y_p = Z_p + \alpha \cdot \bar{Z}_p^{\text{ctx}}.$$

Sharing SE parameters across persons makes SoSE-X *permutation-equivariant* to actor order; averaging confers *scale-invariance* in group size; and initializing $\alpha = 0$ yields an *identity-safe* start that lets the model grow cross-person coupling up to the extent of usefulness. Compared to classic SE, SoSE-X extends attention to both *channel and space* before fusion. It also injects *social context* through a lightweight residual rather than expensive pairwise attention. Empirically, this design provides *context-aware* embeddings that highlight socially informative channels and locations while remaining parameter-efficient. In Section 6.3, we show that ablating SoSE-X causes consistent drops in group-level performance metrics.

## 5.4 INTRA-MODALITY FUSION

To fuse heterogeneous but time-aligned substreams within a modality branch while remaining robust to missing data and compute-efficient, we use *BSC-X*, a mask-aware branch separable mixer that turns multiple feature maps into a compact vector. Given substreams with binary validity masks, GSX performs channel fusion via a $1 \times 1$ projection with channel-wise layer norm and GELU on the concatenation $\mathbf{Z} = [\mathbf{Z}_1; \ldots; \mathbf{Z}_K]$, applies the union mask $\mathbf{M}^\vee = \max_i \mathbf{M}_i$ multiplicatively, and refines local temporal structure with a partial CNXv2 block. The refined map is summarized by two complementary masked statistics over $T$ (a partial mean and a partial generalized mean (GeM)), which are concatenated and layer-normalized, then mapped by a small head to the branch output, i.e., $\mathbf{y} = \text{head}(\text{LN}([\text{Mean}; \text{GeM}]))$.

This design replaces heavy attention or staged gating with a single pointwise fusion, a lightweight temporal refinement, and dual masked poolings that balance stability (mean) and saliency (GeM). GSX is *mask robust* (union masks and partial operations prevent leakage from invalid regions),

*order-agnostic* across substreams (channel concat + $1\times1$ mixing), and *efficient*—compute is dominated by one $1\times1$ projection and global reductions. Section 6.3 demonstrates that BSC-X outperforms simple concatenation and mean pooling under temporal jitter and synthetic noise corruptions.

## 5.5 Cross-Modality Fusion

At the multimodal apex, each modality $i \in \{1, \ldots, M\}$ yields a single vector $\mathbf{Z}_i \in \mathbb{R}^{D_i}$. We introduce *Gated Low-Rank Cross Fusion (GLR-X)* to fuse these vectors with two lightweight components: (i) an allocation gate that forms an availability-aware convex mixture, and (ii) a low-rank bilinear mixer (MLB-style (Kim et al., 2016)) that captures pairwise cross-modal dependencies at low cost. Formally, we (a) project each input into a common space $\mathbf{H}_i = \mathbf{P}_i \operatorname{LN}(\mathbf{Z}_i) \in \mathbb{R}^{D_h}$, (b) compute sample-wise allocation weights $\mathbf{g} \in \Delta^{M-1}$ with an $\varepsilon$-floor tied to modality availability, (c) accumulate a rank-$R$ multilinear signal over any possible pairs, and (d) combine the gated sum and the pairwise term via layer norm and an MLP head to obtain $\mathbf{y} \in \mathbb{R}^{D_{\text{out}}}$.

Let $\pi \in \{0,1\}^M$ denote an availability vector (1 if a modality is present; at training we optionally apply modality dropout to stochastically set some entries to 0). A shared tiny MLP produces allocation logits $\ell_i = \operatorname{MLP}(\mathbf{H}_i)$. We form a masked softmax and an availability-uniform baseline $\mathbf{w} = \operatorname{softmax}(\boldsymbol{\ell} \odot \boldsymbol{\pi} + (-\infty)(\mathbf{1} - \boldsymbol{\pi}))$, $\mathbf{u}_{\text{avail}} = \frac{\boldsymbol{\pi}}{\max\{1, \sum_j \pi_j\}}$, and stabilize with an $\varepsilon$-floor: $\mathbf{g} = (1-\varepsilon)\,\mathbf{w} + \varepsilon\,\mathbf{u}_{\text{avail}}, \mathbf{s} = \sum_{i=1}^{M} g_i \,\mathbf{h}_i$. To encode pairwise interactions, each modality is projected to a rank-$R$ latent via $\boldsymbol{\alpha}_i = \mathbf{A}_i \mathbf{h}_i \in \mathbb{R}^R$, $\mathbf{A}_i \in \mathbb{R}^{R \times D_h}$, and active pairs are accumulated with Hadamard products in an MLB-style fashion (Kim et al., 2016): $\mathbf{p} = \sum_{1 \le i < j \le M} \pi_i \pi_j (\boldsymbol{\alpha}_i \odot \boldsymbol{\alpha}_j) \in \mathbb{R}^R$. Let $m = \sum_i \pi_i$ and $N_{\text{pairs}} = \max(1, \frac{m(m-1)}{2})$; with $|\mathcal{P}| = \frac{M(M-1)}{2}$, we keep the scale invariant under missingness by $\widehat{\mathbf{p}} = \frac{|\mathcal{P}|}{N_{\text{pairs}}} \mathbf{p}, \mathbf{r} = \beta\,\mathbf{W}_{\text{pair}} \widehat{\mathbf{p}} \in \mathbb{R}^{D_h}, \mathbf{W}_{\text{pair}} \in \mathbb{R}^{D_h \times R}$. The fused representation and head are $\mathbf{u} = \operatorname{LN}(\mathbf{s} + \mathbf{r}), \mathbf{y} = \operatorname{MLP}(\operatorname{Dropout}(\mathbf{u}))$.

GLR-X is permutation-equivariant over modalities (shared gate, symmetric pairs) and robust to missingness (masked softmax, $\varepsilon$-floor, pair-count normalization). It separates unary and pairwise evidence with a tunable $\beta$, aiding interpretability. Compared to *Concat+MLP*, GLR-X explicitly learns allocation and multiplicative interactions; compared to a *gated sum only*, it recovers cross-modal cues via low-rank products; compared to *full bilinear/tensor* fusion, it keeps parameters and compute small with rank-$R$ factors; and unlike *cross-attention at the vector apex*, it avoids quadratic scoring while retaining most of the benefit in this compressed regime.

We cite MLB for the *low-rank bilinear* building block between *two* vectors (Kim et al., 2016), i.e., $(Ax) \odot (By)$, but GLR-X extends and integrates it in four key ways: (1) Multi-modality $M \geq 2$ via a symmetric sum over all active pairs $\sum_{i<j}$ with pair-count normalization to keep interaction scale stable when some modalities are missing; (2) an availability-aware allocation gate with an $\varepsilon$-floor that explicitly mixes modalities before interactions, preventing dead experts and calibrating contributions; (3) a decoupled unary vs. pairwise pathway–a gated sum $\mathbf{s}$ (unary evidence) and a scaled MLB-style term $\mathbf{r}$ with an interpretable global coefficient $\beta$–combined for identifiability and stable optimization; and (4) permutation equivariance across modalities through shared gating and a symmetric pair accumulator. In contrast, vanilla MLB is defined for two inputs, does not address availability/missingness, and typically the low-rank product feeds the classifier directly. Our ablations in Section 6.3 show that GLR-X yields better performance than additive cross-modal fusion under modality dropout and conflicting cues.

## 5.6 Classification Heads

SyntalNet employs two classification heads (individual- and group-level) to classify corresponding constructs of cohesion, following a shared-trunk, per-head adapter, and a lightweight classifier design. Each head predicts over a task-specific label set.

The trunk maps fused representations into a task-shared space: for individual-level heads, this is a shallow residual MLP; for group-level heads, a DeepSets-style trunk ensures permutation invariance across members. Head-specific adapters then refine the shared representation through lightweight LoRA–FiLM modules, followed by a cosine–prototype classifier. The classifier interpolates between parametric class weights and EMA-updated prototypes, with a warm-up schedule and per-class fu-

sion weights providing stable training. A full mathematical specification of the trunk, adapter, and cosine–prototype classifier is provided in the Appendix A.6.

# 6 RESULTS

## 6.1 EXPERIMENTAL SETUP

**Tasks.** We evaluate 3-way trend classification (decrease/stable/increase) for five cohesion constructs: group synchrony, group confidence, group interaction phase, and individual engagement and leadership. Labels are derived by forming adjacent clip-pairs and comparing the temporally smoothed posterior for each construct across the pair.

**Training Details.** We optimize a class-balanced focal loss to address the three-way imbalance; AdamW with weight decay 0.002; warm-up for initial 8% then cosine LR schedule (base 1e-5, peak 3e-4); batch size 12 (3 people per sample), gradient accumulation 4; global-norm clipping 1. We use an RTX A6000 (see Appendix A.7 for more details).

**Evaluation Protocol.** Unless otherwise noted, we report results under 5-fold cross-validation at the clip level. We report mean ± standard deviation across folds for all metrics. In Section 6.3 we additionally report a leave-one-group-out (LOGO) protocol in which all clips from one triad are held out for evaluation and models are optionally fine-tuned on a subset of that triad.

**Baseline Formulation.** We compare our model (SyntalNet) against the following baselines, which are capacity-matched and tuned independently. We select them because they are strong, widely used models in time series estimation and categorical classifications for human states (Pramerdorfer & Kampel, 2016; Quddus et al., 2021; Zhu et al., 2025).

- **Logistic Regression.** Operates on pooled synchrony features, including mutual attention, facial action unit synchrony, shared gaze, etc., and serves as an explicit "aggregation fallacy" control.
- **Temporal BiLSTM.** Mask-aware per-modality bidirectional, multi-layer LSTM encoders produce per-branch embeddings averaged across modalities and projected, branch features are concatenated,a shared 1-layer MLP plus per-head linear classifiers produce logits.
- **Temporal CNN.** Mask-aware per-modality temporal CNNs; within each branch the modality encodings are averaged and projected, branch features are concatenated, a shared 1-layer MLP plus per-head linear classifiers produce individual logits and group logits.
- **VLM baseline.** Frame-sample clips and prompt a VLM 5 times to output majority vote deltas. We also computed the standard deviations of the 5 VLM predictions averaged on each target. These standard deviations are 0.160, 0.124, 0.198, 0.198, and 0.179 for individual engagement, individual leadership, group synchronization, group confidence, and interaction phase, respectively. The prompt and example outputs can be found in A.8.

## 6.2 QUANTITATIVE RESULTS

Since the label distribution is highly imbalanced, we report macro AUPRC and macro F1 as our primary metrics. Table 1 summarizes these scores for all five constructs at both individual and group levels; accuracy and AUROC are deferred to Appendix A.1 (Table 2).

Across all constructs, SyntalNet consistently outperforms the temporal baselines, with the largest margins on group-level states. In particular, it yields substantial gains over the Temporal CNN and BiLSTM, especially for constructs that rely on cross-participant cues and cross-modal evidence. This pattern is consistent with the design of SyntalNet, which incorporates explicit cross-person fusion (SoSE-X) and low-rank, mask-aware cross-modal fusion (GLR-X), while the temporal baselines operate with additive pooling and lack availability-aware gating.

VLMs are competitive on individual states (e.g., Leadership macro F1 $\approx 0.56$) but remain close to chance on group states (0.36–0.39 F1), whereas SyntalNet reaches 0.65–0.67 F1 on the same tasks (an absolute gain of roughly 0.28). This suggests that inferring latent group dynamics from sparse frames and transcripts remains challenging even for large pretrained VLMs. Finally, logistic regression on pooled synchronized features behaves almost like a constant predictor on both individual and group states, underscoring that simple pooled hand-crafted features are insufficient and that explicit, dynamic modeling of cross-participant and cross-modal interactions is necessary.

Table 1: Quantitative results on Syntality: macro AUPRC and macro F1 averaged over 5-fold CV. SyntalNet outperforms all baselines, with the largest improvements on group-level constructs. Full metrics (accuracy, AUROC) are deferred to Appendix A.1.

(a) AUPRC

| Model | Indiv Eng. | Indiv Lead. | Group Sync. | Group Conf. | Interact. Phase |
|---|---|---|---|---|---|
| GPT-5 | 0.468 | 0.495 | 0.351 | 0.363 | 0.427 |
| LogReg | 0.374 | 0.357 | 0.357 | 0.367 | 0.340 |
| LSTM | 0.471 | 0.426 | 0.487 | 0.437 | 0.480 |
| Temporal CNN | 0.514 | 0.462 | 0.539 | 0.530 | 0.570 |
| SyntalNet (Ours) | 0.690 | 0.638 | 0.697 | 0.668 | 0.670 |

(b) F1 Score

| Model | Indiv Eng. | Indiv Lead. | Group Sync. | Group Conf. | Interact. Phase |
|---|---|---|---|---|---|
| GPT-5 | 0.418 | 0.461 | 0.358 | 0.379 | 0.399 |
| LogReg | 0.366 | 0.351 | 0.371 | 0.346 | 0.392 |
| LSTM | 0.424 | 0.414 | 0.521 | 0.436 | 0.441 |
| Temporal CNN | 0.506 | 0.435 | 0.547 | 0.530 | 0.535 |
| SyntalNet (Ours) | 0.676 | 0.634 | 0.645 | 0.672 | 0.662 |

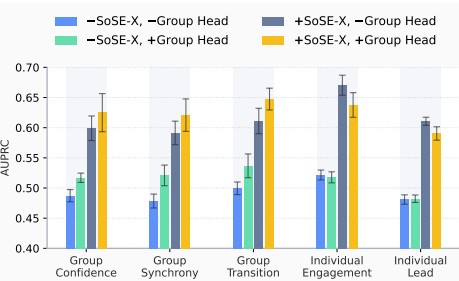

Figure 2: **Ablation of cross-participant fusion.** Macro AUPRC per construct when toggling SoSE-X in the encoder and the DeepSets-style group head. The purely additive model (no SoSE-X, no group head) performs worst; adding only a group head yields modest gains; introducing SoSE-X in the encoder yields a large improvement, and combining SoSE-X with the group head yields the best performance across all group constructs. Error bars show mean $\pm$ std over folds.

6.3 ABLATIONS AND EXPERIMENTS

We next probe the contribution of each fusion module in SyntalNet and the transferability of the learned representations. Unless stated otherwise, all ablations are run on the 5-fold random split protocol and we report macro AUPRC averaged over the corresponding constructs.

**Cross-participant fusion (SoSE-X).** To test our central hypothesis that additive pooling of independently encoded participants is insufficient, we toggle (i) the SoSE-X encoder block, which injects cross-participant context before temporal aggregation, and (ii) the DeepSets-style group head, which only mixes participants at the prediction stage. This yields four models: a purely additive baseline (no SoSE-X, no group head), a *head-only* variant (no SoSE-X, with group head), an *encoder-only* variant (with SoSE-X, no group head), and the full model (with both). As shown in Figure 2, the purely additive baseline performs worst across all constructs. Adding only a group head yields modest gains, especially on group-level constructs, but leaves individual engagement/lead almost unchanged. Introducing SoSE-X in the encoder yields a much larger jump in AUPRC (roughly $+0.11$ AUPRC on average for group confidence/synchrony/transition), and combining SoSE-X with the group head provides the highest performance on all group construct. This pattern directly supports our claim that explicit cross-participant fusion *within* the encoder is the main driver of improved group-state prediction, with the group head providing an additional but smaller benefit.

**Intra-modality robustness (BSC-X).** To stress-test branch-level fusion, we inject three families of corruption on the three branches: temporal misalignment jitter, additive feature noise, and temporal band masking. We compare BSC-X against two alternatives: channel-wise mean pooling and Concat+MLP. Results in Appendix A.1 (Figure 4) show that while all models degrade as corruption increases, BSC-X consistently attains higher mean AUPRC and exhibits a slow performance drop.

**Cross-modal robustness (GLR-X).** We next probe cross-modal fusion by corrupting individual modalities with increasing Gaussian noise while keeping the others intact. We compare GLR-X against mean pooling and Concat+MLP at the representation level. As detailed in Appendix A.1 (Figure 5), GLR-X matches or exceeds the baselines at low noise levels and clearly dominates when

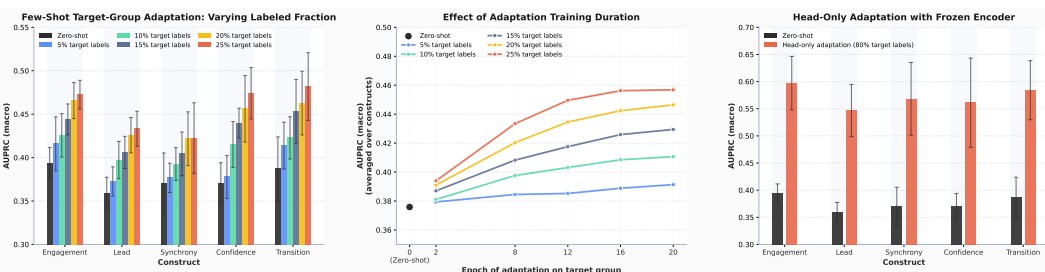

Figure 3: **Transfer to unseen teams.** *Left:* LOGO macro AUPRC per construct as a function of the fraction of labeled clips from the held-out target group used for adaptation. Even 5–10% labeled data yields noticeable gains over the zero-shot baseline, and performance continues to improve up to 25%. *Middle:* Average macro AUPRC (over constructs) versus number of adaptation epochs, showing rapid improvement in the first 10–12 epochs and saturation thereafter. *Right:* Head-only adaptation with the encoder frozen but using 80% labeled target data substantially outperforms zero-shot, indicating that the pretrained encoder learns transferable social representations and that most of the remaining gap is due to re-estimating decision boundaries.

a single modality becomes heavily corrupted. This suggests that the availability-aware gate and low-rank pairwise interactions help the model reweight modalities adaptively.

**Out-of-distribution adaptation to unseen teams.** To evaluate transferability across teams, we run leave-one-group-out (LOGO) experiments where SyntalNet is trained on 9 triads and evaluated on the 10th. We first assess a strict *zero-shot* setting with no target-group labels. We then progressively reveal a small fraction $f \in \{5, 10, 15, 20, 25\}\%$ of labeled clips from the held-out group and fine-tune the entire model for a fixed number of epochs. As shown in Figure 3, zero-shot performance is modest across constructs (AUPRC around 0.35–0.39), but even 5–10% labeled data yields noticeable gains, and with 25% labels the model recovers roughly 0.08–0.11 AUPRC across all 5 constructs. The middle panel of Figure 3 tracks AUPRC as a function of adaptation epochs, showing that most of the improvement occurs within the first 10–12 epochs, after which performance saturates, suggesting the encoder adapts quickly to the idiosyncrasies of a new team. Finally, we test a *head-only* adaptation regime where we freeze the encoder and fine-tune only the individual and group classification heads using 80% of the target-group labels. The right panel of Figure 3 shows a large gap between zero-shot and head-only adaptation (over 0.15 AUPRC), indicating that the pretrained encoder already learns useful, transferable representations and that much of the remaining error stems from misaligned decision boundaries rather than missing social cues. Together, these results suggest that SyntalNet is a promising backbone for few-shot adaptation to new teams.

## 7 LIMITATIONS AND CONCLUSIONS

Our study has limitations that motivate future work. Syntality currently covers $\approx 170$ minutes of interaction in a single collaborative task with fixed triads, limiting diversity in tasks and team structures. Labels are derived from observer ratings rather than first-person reports, reflecting the difficulty of collecting reliable self-assessments at scale. These constraints point toward natural extensions: larger and more varied datasets, broader task domains, and hybrid annotation schemes that combine observer judgments with participant self-reports.

Despite these constraints, the work makes several contributions. We introduce **SyntalNet**, a multimodal architecture explicitly designed to model cross-participant dependencies, and **Syntality**, a benchmark of individual and group trend labels in triadic CPS. Our findings show that emergent team states cannot be reduced to pooled individual signals, underscoring the need for relational modeling in socially aware AI. SyntalNet significantly outperforms baselines, including visual language model, on predicting state changes across synchrony, confidence, interaction phase, engagement, and leadership. Together, these contributions advance the foundation for AI systems that can sense, and adapt to the evolving social context of human teams.

## 8 REPRODUCIBILITY AND ETHICS STATEMENTS

We release the Syntality Dataset with raw and processed audiovisual, skeleton, facial, dialogue, and acoustic features, as well as the full annotation pipeline and derived labels. All preprocessing scripts, model code, and experiment configurations will be made publicly available upon acceptance. This enables full reproduction of our results, including dataset statistics, baselines, and training results.

The Syntality dataset is based on a prior publicly released group problem-solving dataset. We augmented it with additional annotations obtained via the Prolific platform. As this activity did not involve collecting personal or sensitive data, it does not constitute human-subjects research and falls outside IRB oversight.

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

Table 2: **Full metrics for main comparison.** Extension of Table 1 including macro Accuracy and macro AUROC in addition to macro AUPRC and macro F1. Numbers are reported as mean $\pm$ std over 5 folds.

(a) Accuracy

| Model | Indiv Eng. | Indiv Lead. | Group Sync. | Group Conf. | Interact. Phase |
|---|---|---|---|---|---|
| GPT-5 | 0.420 | 0.461 | 0.356 | 0.376 | 0.365 |
| LogReg | 0.375 | 0.373 | 0.379 | 0.354 | 0.388 |
| LSTM | 0.412 | 0.405 | 0.517 | 0.433 | 0.441 |
| Temporal CNN | 0.507 | 0.428 | 0.569 | 0.553 | 0.547 |
| SyntalNet (Ours) | 0.700 | 0.650 | 0.639 | 0.682 | 0.664 |

(b) F1

| Model | Indiv Eng. | Indiv Lead. | Group Sync. | Group Conf. | Interact. Phase |
|---|---|---|---|---|---|
| GPT-5 | 0.418 | 0.461 | 0.358 | 0.379 | 0.399 |
| LogReg | 0.366 | 0.351 | 0.371 | 0.346 | 0.392 |
| LSTM | 0.424 | 0.414 | 0.521 | 0.436 | 0.441 |
| Temporal CNN | 0.506 | 0.435 | 0.547 | 0.530 | 0.535 |
| SyntalNet (Ours) | 0.676 | 0.634 | 0.645 | 0.672 | 0.662 |

(c) AUPRC

| Model | Indiv Eng. | Indiv Lead. | Group Sync. | Group Conf. | Interact. Phase |
|---|---|---|---|---|---|
| GPT-5 | 0.468 | 0.495 | 0.351 | 0.363 | 0.427 |
| LogReg | 0.374 | 0.357 | 0.357 | 0.367 | 0.340 |
| LSTM | 0.471 | 0.426 | 0.487 | 0.437 | 0.480 |
| Temporal CNN | 0.514 | 0.462 | 0.539 | 0.530 | 0.570 |
| SyntalNet (Ours) | 0.690 | 0.638 | 0.697 | 0.668 | 0.670 |

(d) AUROC

| Model | Indiv Eng. | Indiv Lead. | Group Sync. | Group Conf. | Interact. Phase |
|---|---|---|---|---|---|
| GPT-5 | 0.562 | 0.594 | 0.488 | 0.498 | 0.523 |
| LogReg | 0.544 | 0.510 | 0.507 | 0.521 | 0.494 |
| LSTM | 0.695 | 0.634 | 0.700 | 0.646 | 0.666 |
| Temporal CNN | 0.768 | 0.687 | 0.770 | 0.743 | 0.758 |
| SyntalNet (Ours) | 0.880 | 0.826 | 0.837 | 0.835 | 0.845 |

# A    APPENDIX

## A.1    ADDITIONAL RESULTS

This section provides complementary quantitative results and extended ablations that could not be included in the main text due to space. Throughout, we use the same training and evaluation protocol as in Sections 6.2 and 6.3.

### A.1.1    FULL METRIC SUITE FOR MAIN COMPARISON

Table 2 extends Table 1 by reporting balanced Accuracy and macro AUROC in addition to macro AUPRC and macro F1 for all models and constructs.[1] As in the main table, SyntalNet achieves the best performance on every metric, with consistent margins over all baselines. Importantly, the relative ranking of models is essentially unchanged across AUPRC, F1, AUROC, and Accuracy.

### A.1.2    ADDITIONAL ROBUSTNESS ABLATIONS FOR BSC-X AND GLR-X

For completeness, we also report extended robustness analyses for the multichannel fusion (BSC-X) and multimodal fusion (GLR-X) modules that were summarized only briefly in the main paper.

**BSC-X under temporal corruptions.**    Figure 4 extends the multichannel fusion ablation by probing three types of temporal corruptions applied to the per-participant streams at test time: (i) *misalignment jitter*, where we randomly shift each channel independently by up to $\pm k$ time steps; (ii) *feature noise*, where we add zero-mean Gaussian noise with variance proportional to the empirical variance of each channel; and (iii) *temporal band masking*, where we drop contiguous blocks of frames. We compare BSC-X to two alternatives that share the same encoder: a uniform "mean pooling" over channels and a "Concat+Proj" variant that concatenates all channels and applies a learned linear projection. Across all three perturbations, BSC-X consistently attains higher macro AUPRC and degrades gracefully as the corruption level increases.

**GLR-X under modality noise and dropout.**    Figure 5 complements the main multimodal fusion ablation by separating performance on individual-level versus group-level targets. After pretraining the full model, we replace the original fusion block with (i) mean pooling and (ii) a Concat+MLP baseline, reinitialize the fusion parameters, and fine-tune for 30 epochs while randomly corrupting modalities with Gaussian noise and modality dropout. We then evaluate macro AUPRC as a function of noise scale. GLR-X consistently matches or exceeds the strongest baseline at low noise levels

---

[1]All metrics are computed per construct and then averaged over the validation folds.

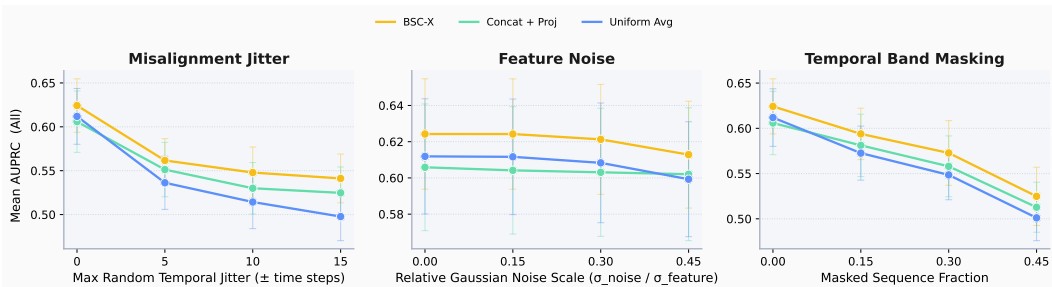

Figure 4: **Robustness of intra-modality fusion (BSC-X) to temporal corruptions.** Mean macro AUPRC across all constructs under increasing levels of (left) misalignment jitter, (middle) additive feature noise, and (right) temporal band masking. BSC-X (yellow) consistently outperforms uniform averaging and the Concat+Proj baseline and exhibits slower performance degradation as corruptions intensify.

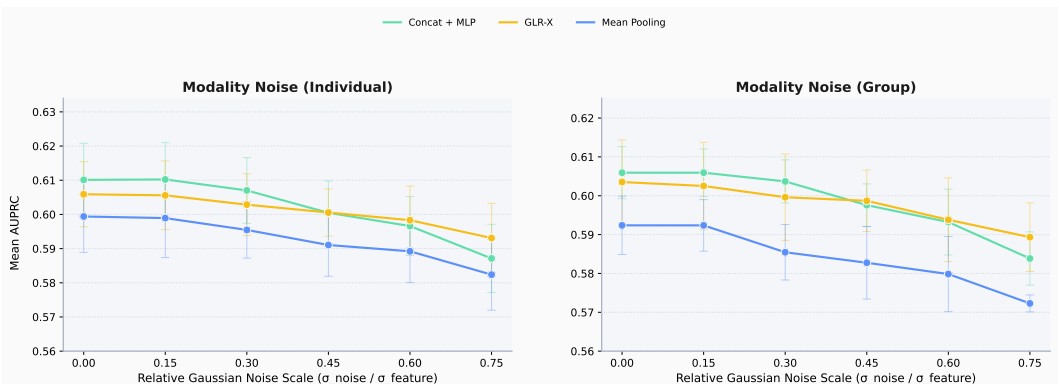

Figure 5: **Multimodal robustness of GLR-X.** Macro AUPRC (averaged over constructs) versus relative Gaussian noise scale for (left) individual-level constructs and (right) group-level constructs. GLR-X (yellow) is consistently more robust to modality noise and dropout than mean pooling and Concat+MLP.

and maintains a clear advantage as noise increases, especially for group-level constructs. The gap between GLR-X and the alternatives widens in the high-noise regime, indicating that low-rank, mask-aware interactions help the model preserve useful cross-modal structure rather than overfitting to any single modality.

## A.2 DATASET: TASK

The task equipment includes 6 blocks (of varying weight, size, and color), a balance scale, a worksheet, and a computer with a survey where participants submit their answers. An example scene of a triad together solving a task is shown in Fig. 6. Each triad first uses a balance scale to infer the weights of five blocks given the first block's weight (10g). As each weight is determined, the block is placed on the worksheet in the cell corresponding to that value. In the next phase, participants receive a new block and, without using the scale, deduce its weight from the pattern observed in the initial set. Finally, they predict the weight of the next hypothetical block and explain their reasoning. After each stage, each triad submits their answers via the survey form. There are a total of 10 triads, and their audio-visual signals are available. The total data length is 170 minutes. Each triad took an average of 17.0 minutes to complete the session, with session time varying from 9 to 34 minutes.

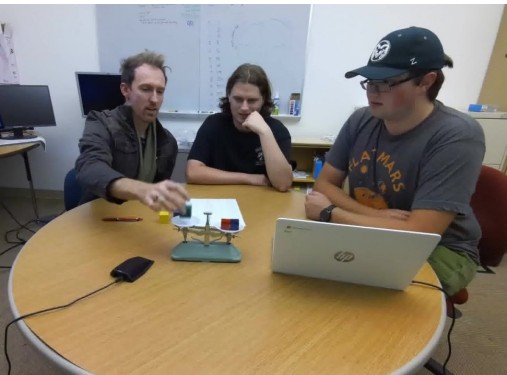

Figure 6: Task Scene

## A.3 DATASET: ANNOTATION QUESTIONS

Table 3: Annotation Questions

| Category | Question |
| --- | --- |
| Cohesion | Individual efforts are synchronized within the team. (1-5: strongly disagree to strongly agree) |
| Confidence | The team has confidence in itself. (1-5: strongly disagree to strongly agree) |
| Interaction Phase | Which interaction phase best describes the group's current activity? (among Starting, Conflict, Setting, Performing, Wrapping up) |
| Engagement | How engaged is each person? (1-3: passive to active) |
| Leadership | What are the social roles of these people? (1-3: follower to leader) |

## A.4 DATASET: IMPLEMENTATION DETAILS

### A.4.1 VIDEO

For each group we process three synchronized viewpoint videos (sub1, master, sub2), each predominantly centered on a distinct participant. Per view, persons are detected with YOLO and temporally associated with ByteTrack, yielding per-frame boxes

$$b_t = \left( x_1^{(t)}, \, y_1^{(t)}, \, x_2^{(t)}, \, y_2^{(t)} \right)$$

and a stable track ID. For a participant $p$ we aggregate all boxes $\{b_t\}_p$ and define a robust, time-invariant crop by the componentwise quantiles

$$\tilde{x}_1 = Q_{0.2}\big(\{x_1^{(t)}\}_p\big), \quad \tilde{y}_1 = Q_{0.2}\big(\{y_1^{(t)}\}_p\big), \quad \tilde{x}_2 = Q_{0.8}\big(\{x_2^{(t)}\}_p\big), \quad \tilde{y}_2 = Q_{0.8}\big(\{y_2^{(t)}\}_p\big),$$

where $Q_q(\cdot)$ denotes the empirical $q$-quantile. This process suppresses outliers and short-lived jitter. Frames are then cropped to $\tilde{b} = (\tilde{x}_1, \tilde{y}_1, \tilde{x}_2, \tilde{y}_2)$, resized with aspect-ratio preservation, and padded to $224 \times 224$. The resultant person-centric streams retain native timing; snippet generation (when used in experiments) slides a window of length $D$ seconds with stride $S$ seconds over these crops without altering geometry.

### A.4.2 BODY KEYPOINTS

Each scene contains three views of Kinect skeleton streams. We synchronize by timestamp and map participant identities across views via known track IDs. We re-express all joint coordinates to a consistent camera-centric system by swapping Y/Z, negating Z (right-handed convention), and root-centering at the pelvis to remove global translation. Because participants are seated at a round table with frequent lower-limb occlusions, we retain upper-body joints for training/evaluation. For each frame and participant, we fuse evidence across views to produce a single 3D, root-relative skeleton. If a participant is visible in the primary view, we take those joint coordinates as the initial

estimate. If the primary view is missing or partial but other views are available, we import the best alternate view and perform rigid alignment via Kabsch rotation estimated on upper-body anchors. The aligned skeleton is then re-centered at the pelvis. If a joint is fully occluded, we do not impute such a joint and leave it missing. To accommodate use cases preferring temporally stabilized inputs, we include hybrid forward/backward filling at the per-joint level for short spans only and smoothing with Savitzky–Golay filtering with window=7 and polyorder=2 applied to joint trajectories.

Each scene provides three Kinect skeleton streams. We first synchronize frames by the common timestamps and map identities across views using known track IDs. All joint coordinates are expressed in a common, camera-agnostic body frame. Formally, if a raw joint is given as $p = (x, y, z) \in \mathbb{R}^3$ in the original Kinect coordinates, we apply an axis permutation and sign flip as $(x, y, z) \mapsto (x, z, -y)$ to restore a right-handed convention and consistent vertical axis across views, and then re-express joints root-relatively by subtracting the pelvis position $p_{\text{pelvis}}$ as $\hat{p}_j = p_j - p_{\text{pelvis}}$.

For each participant and frame we select their *primary view*. If the primary view provides a skeleton in a frame, its joint set $\{\hat{p}_j\}_{j=1}^J$ is used. If the primary view is entirely missing in that frame but an alternate view contains the participant, we import that skeleton and rigidly align it to the primary camera's frame. Alignment is performed with a Kabsch solution using a set of upper-body anchors $J_{\text{UB}}$ (pelvis, spine, clavicles/shoulders, elbows/wrists/hands, head, and face-proximal joints). Let

$$P, Q \in \mathbb{R}^{|J_{\text{UB}}| \times 3}$$

denote the centered anchor matrices from the primary and fallback views, respectively. The optimal rigid rotation mapping $Q$ (fallback) to $P$ (primary) is

$$R^\star = \arg\min_{R \in SO(3)} \| RQ - P \|_F, \qquad H = Q^\top P, \quad U\Sigma V^\top = \text{SVD}(H), \quad R^\star = VU^\top,$$

where $SO(3) = \{R \in \mathbb{R}^{3 \times 3} : R^\top R = I, \det(R) = 1\}$ denotes the group of 3D rotation matrices. The standard reflection fix is applied to ensure $\det(R^\star) > 0$. The rotated fallback skeleton is then re-centered at the pelvis. The estimate $R^\star$ is computed from the most recent past frame where both views observe the participant and applied to the current fallback frame.

We keep all joints in the output, although only upper-body joints are later used for training and evaluation due to frequent lower-limb occlusions around the table. Joints that are missing across all views are not imputed. For temporally stabilized inputs, we provide a variant where short contiguous gaps are interpolated by a symmetric forward/backward fill on each coordinate trajectory, followed by smoothing with a Savitzky–Golay filter (window size 7, polynomial order 2) applied independently per joint coordinate. This reduces sensor jitter while preserving local motion trends.

### A.4.3 FACE KEYPOINTS, HEAD, GAZE

Facial signals are extracted with OpenFace, yielding per-frame 68-point 2D and 3D landmarks, global head pose coordinate and rotation, and averaged gaze angles. We associate detections to participants using the video tracks from Section A.4.1. For a given frame, we form a landmark-derived face box $f = (\min X, \min Y, \max X, \max Y)$ and test containment $f \subseteq \tilde{b}_p$ within the participant's box. If exactly one detection is contained, we assign it; if multiple are contained, we select the highest OpenFace confidence; otherwise we assign the detection with maximal IoU with $\tilde{b}_p$.

To reduce camera and scale dependence, landmarks are canonicalized independently per person-frame. Let $L = \{(x_i, y_i)\}_{i=1}^{68}$ (2D) and $L^{3D} = \{(X_i, Y_i, Z_i)\}_{i=1}^{68}$ (3D). We subtract the centroid $\bar{\ell}$ of the set and divide by a face reference length

$$s = \max(\|\ell_0 - \ell_{16}\|, \|\ell_8 - \ell_{27}\|),$$

where $\ell_k$ denotes the $k$-th landmark in the standard indexation, and the tuple represents the jaw width and chin–nose ridge, respectively. The normalized landmarks are $\tilde{\ell}_i = (\ell_i - \bar{\ell})/s$, and the same centering and scaling are applied to $L^{3D}$. Similar to pose data, we also provide stabilized variants formed by short-gap filling and Savitzky–Golay smoothing (window = 7, polyorder = 2) on the normalized 2D/3D landmark trajectories and on gaze angles. Head-pose parameters are retained in the camera frame to preserve interpretability of absolute head motion.

### A.4.4 TURN-TAKING

Given the automatic transcripts from Google Cloud ASR and Whisper, we manually corrected both text and segment boundaries, where ASR merged multiple turns. The moderator turns are removed. The final table is sorted by start time and, for each utterance $u_i$, contains participant label $p_i \in \{1, 2, 3\}$, start $s_i$, end $e_i$, duration $d_i = e_i - s_i$, and text $w_i$.

$\not\Vdash[\cdot]$A normalized progress indicator (Turn Position) used downstream is $i/N$, where $i$ is the utterance index and $N$ the total per session. Furthermore, let $\{u_i\}_{i=1}^{N}$ be the utterances after sorting and moderator removal. Define the running prior end time as

$$m_i = \max_{j<i} e_j \quad (m_1 = 0),$$

the pause before utterance $i$ as

$$\pi_i = \max\{0, \, s_i - m_i\},$$

and a speaker change flag as $\chi_i = \mathbb{1}[p_i \neq p_{i-1}]$ for $i > 1$. An overlap occurs when $\omega_i = \mathbb{1}[s_i < m_i] = 1$; the "host" utterance is is the previous utterance with the latest end time defined as $h_i = \arg\max_{j<i} e_j$.

---

### LLM Prompting for Bachchannel Detection

**System Message:**

```
I want you to act as a conversation analyst that determines whether a brief
interruption in conversation is actually a backchannel or just a butt-in. For each
interaction, I will send you input in JSON format with the following structure:
{
"host utterance": {
    "text": "...The ongoing sentence spoken by the main speaker...",
    "start_time": 0.00,
    "end_time": 2.40
},
"interruption": {
    "text": "...the brief interjection from another speaker...",
    "start_time": 1.85,
    "end_time": 2.10
}
}
Your task is to determine if the interjection is backchanneling in the context of
conversational dynamics. Backchanneling includes brief verbal signals that indicate
attention, understanding, agreement, or encouragement without attempting to take
over the turn. For each pair I send, respond in the following JSON format:
{
    "analysis": "...Your reasoning about the interruption in one sentence - include
                details about form, timing, function, etc. ...",
    "backchannel": "Yes" or "No"
}
```

**User Payload (template):**

```
{
  "host utterance": {
    "text": "<HOST_TEXT>",
    "start_time": <HOST_START_SECONDS>,
    "end_time": <HOST_END_SECONDS>
  },
  "interruption": {
    "text": "<INTERRUPT_TEXT>",
    "start_time": <INT_START_SECONDS>,
    "end_time": <INT_END_SECONDS>
  }
}
```

**Expected model output (schema):**

```
{
  "analysis": "<one-sentence rationale>",
  "backchannel": "Yes" | "No"
}
```

Overlaps are labeled:

$$\text{floor-taking } (\phi_i) \;=\; \mathbb{1}[\omega_i \wedge e_i \geq m_i], \qquad \text{butting-in } (\beta_i) \;=\; \mathbb{1}[\omega_i \wedge e_i < m_i].$$

To distinguish backchannels from butting-ins, each pair $(h_i, i)$ with $\beta_i = 1$ is adjudicated using the OpenAI Responses API (model gpt-4.1, temperature=1, top_p=1), with a constrained JSON protocol. To force the model to reason before deciding, it was instructed to return analysis first, followed by the decision. The model receives a fixed *system* instruction and a per-pair *user* payload; on JSON parse failure, we attempt one retry while unresolved indices are logged for manual labeling. The required output is a single JSON object with fields "analysis" and "backchannel". In the event of a decision in favor of backchannel, we set $\beta_i \leftarrow 0$ and $bc_i \leftarrow 1$, and store the rationale. A single retry is used on JSON parse failure; unresolved indices are logged for manual labeling. Below are the system and user messages employed.

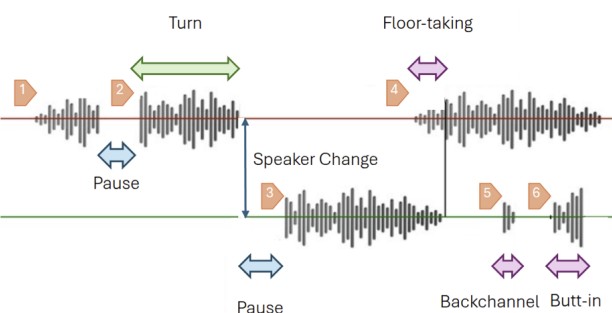

Figure 7: Overview of conversational turn-taking events.

### A.4.5 UTTERANCE

**Text embeddings and sparsification.** Given a transcript $x$, we form a lightweight prompt $p(x)$ and tokenize it into a sequence $\{w_t\}_{t=1}^{T}$ with left padding and attention mask $m \in \{0,1\}^T$. A decoder-only LLM (Llama-3.3-70B-Instruct) produces final-layer hidden states $H^{(L)} \in \mathbb{R}^{T \times d}$; we take the index of the last real token $t^\star = \sum_{t=1}^{T} m_t$ and define the sentence embedding as the *last-token* (EOS) vector

$$\mathbf{e}(x) \triangleq H_{t^\star}^{(L)} \in \mathbb{R}^d,$$

a standard surrogate for a CLS representation in causal LMs, since $\mathbf{e}(x)$ attends (causally) to all preceding tokens and is the state used by sequence heads in GPT-style models (Muennighoff, 2022). To harmonize $\mathbf{e}(x)$ with lower-level modalities and reduce computation, we compress it using a sparse autoencoder with encoder $f_\phi : \mathbb{R}^d \to \mathbb{R}^k$ and decoder $g_\psi : \mathbb{R}^k \to \mathbb{R}^d$ ($k \ll d$), yielding a latent $\mathbf{z} = f_\phi(\mathbf{e})$ and reconstruction $\hat{\mathbf{e}} = g_\psi(\mathbf{z})$. We train $(\phi, \psi)$ with an $\ell_2$ reconstruction term and a KL sparsity penalty on the (absolute) latent activations,

$$\mathcal{L}(\phi, \psi) = \|\hat{\mathbf{e}} - \mathbf{e}\|_2^2 \; + \; \beta \sum_{j=1}^{k} \mathrm{KL}\big(\rho \,\|\, \hat{\rho}_j\big), \quad \hat{\rho}_j = \mathbb{E}[\, |z_j| \,],$$

which encourages a compact, semantically structured code while preserving task-relevant information (Hinton & Salakhutdinov, 2006). In practice we use mini-batch optimization with left-padding, extract $\mathbf{e}(x)$ by masking the last non-pad position, and store both $\mathbf{z}$ and per-utterance metadata for downstream multimodal fusion.

### A.4.6 ACOUSTICS

Let $x[n]$ be the discrete-time waveform sampled at $F_s$ Hz. We compute features on overlapping frames

$$x_t[m] = x[tH + m]\, w[m], \quad m = 0, \dots, N-1,$$

where $w[m]$ is a window (e.g., Hann), $N$ is the frame length, and $H$ is the hop size.

Define the short-time autocorrelation of frame $x_t$ as

$$R_t[\tau] = \sum_{m=0}^{N-1-\tau} x_t[m]\, x_t[m+\tau], \quad \tau \geq 0.$$

Search lags $\tau \in [\tau_{\min}, \tau_{\max}]$ corresponding to a frequency range $[f_{\max}, f_{\min}]$ with $\tau_{\min} = \left\lfloor \frac{F_s}{f_{\max}} \right\rfloor$ and $\tau_{\max} = \left\lfloor \frac{F_s}{f_{\min}} \right\rfloor$. The pitch period $\hat{\tau}_t$ and fundamental frequency $\hat{f}_{0,t}$ are

$$\hat{\tau}_t = \operatorname*{arg\,max}_{\tau \in [\tau_{\min}, \tau_{\max}]} R_t[\tau], \qquad \hat{f}_{0,t} = \frac{F_s}{\hat{\tau}_t}.$$

(Optionally, use the normalized autocorrelation $R_t[\tau]/R_t[0]$ and apply voicing thresholds.)

Using the same autocorrelation,

$$\mathrm{HNR}_t = 10 \log_{10} \left( \frac{R_t[\hat{\tau}_t]}{R_t[0] - R_t[\hat{\tau}_t]} \right).$$

Here, $R_t[\hat{\tau}_t]$ approximates harmonic power and $R_t[0] - R_t[\hat{\tau}_t]$ approximates noise power.

Compute the $N_{\text{FFT}}$-point DFT of $x_t$:

$$X_t[k] = \sum_{m=0}^{N-1} x_t[m]\, e^{-j2\pi \frac{km}{N_{\text{FFT}}}}, \quad k = 0, \dots, N_{\text{FFT}} - 1,$$

and the power spectrum $P_t[k] = \frac{1}{N_{\text{FFT}}} |X_t[k]|^2$. Apply a bank of $K$ triangular mel filters $\{H_k(f)\}_{k=1}^{K}$:

$$E_{t,k} = \sum_{k'=0}^{N_{\text{FFT}}-1} P_t[k']\, H_k\left(\frac{k' F_s}{N_{\text{FFT}}}\right), \quad k = 1, \dots, K.$$

Take log energies and the type-II DCT to obtain MFCCs $c_{t,n}$ for $n = 0, \dots, M - 1$ (typically $M \leq K$):

$$c_{t,n} = \sum_{k=1}^{K} \log(E_{t,k})\, \cos\left[\frac{\pi n}{K}\left(k - \tfrac{1}{2}\right)\right].$$

(Optional: discard $c_{t,0}$, append $\Delta/\Delta^2$ via regression.)

We use windowed short-time energy

$$E_t = \sum_{m=0}^{N-1} \left(x[tH + m]\, w[m]\right)^2,$$

or equivalently the RMS energy

$$\text{RMS}_t = \sqrt{\frac{1}{N} \sum_{m=0}^{N-1} \left(x[tH + m]\, w[m]\right)^2}.$$

### A.4.7  LABEL DISTRIBUTION

We visualize the label distribution, where the "Same" class are intuitively the majority across all categories.

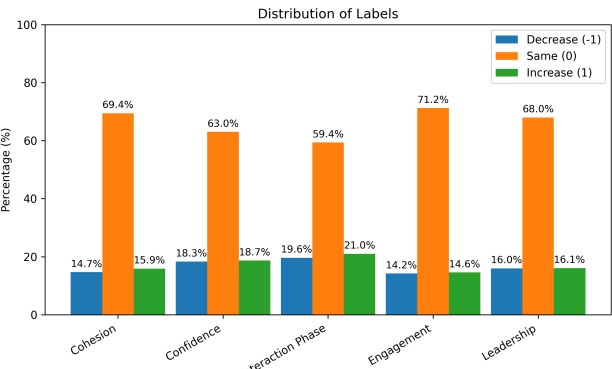

Figure 8: Distribution of Labels

### A.4.8  REPRODUCIBILITY

Unless otherwise stated in the experiments section, all streams described above are finally resampled to 10 Hz to provide a common cadence for multimodal fusion and training. Both raw sources and processed artifacts, together with scripts that implement Sections A.4.1–A.4.3, are released for full reproducibility.

## A.5 STATISTICAL ANALYSIS

We conduct statistical analysis to reveal behavioral patterns. First, we ran a Pearson correlation test between the individual and group states, and found out very low correlation (<0.18) between them. Correlations are especially low between leadership roles and group cohesion, confidence, and inter-action phases, as shown in 9. We train an interpretable baseline by fitting per-target ordinary least squares (OLS) regression model that map individual directional-trend indicators to group trends. We used a 70/30 random split, z-score predictors using training statistics, fit a separate OLS for each target. We summarize the performance with a $R^2$ bar chart, as shown in 10. Our simple aggregation baseline fails to explain the group targets, performing barely above a constant predictor. This lack of linear association indicates that group states are not reducible to pooled individual states. Instead, they depend on relational dependencies and coordinated dynamics.

We further examined the relationship between multimodal behavioral patterns and group states. We extracted some interaction features to represent group interplays, including mutual attention, facial action unit synchronization, and shared gaze. We found that mutual attention is significantly higher when the team confidence increases, perhaps participants are confirming their confidence to each other. When individual engagement decreases, the shared gaze fraction significantly drops, as well as the mutual attention. This pattern indicates that people do not look at shared objects or each other when they are losing interest in the task. Finally, one's facial action units have less synchronization with teammates when their leadership role decreases, potentially revealing conflicts. Details of the synchronized feature extraction, and analysis visualization can be found in 11. These behavioral patterns further validate our hypothesis that individual and group states have strong indicators from interplays between participants in a group interaction, which needs careful modeling.

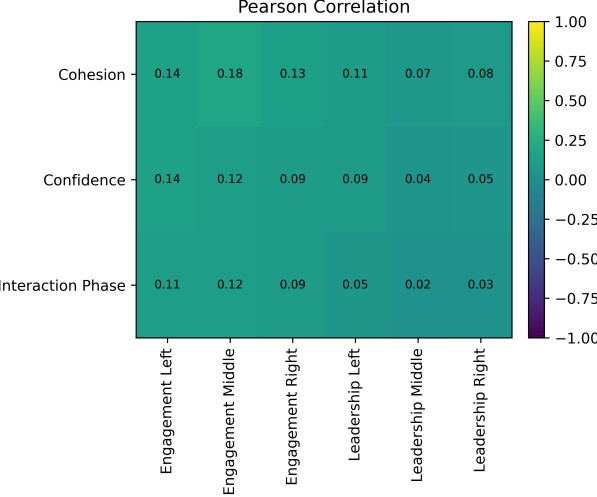

Figure 9: Pearson Correlation

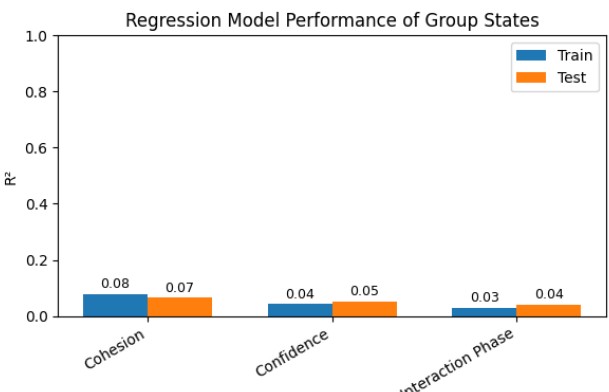

Figure 10: Regression Results

Mutual attention is computed as the fraction of time where at least one person is looking toward another person's head (Reddy, 2005). The action unit synchronization is computed as the mean pairwise correlation across the triad, of all their facial action units (Valstar & Pantic, 2006). The shared gaze measures how often everyone's gaze directions are aligned using the mean resultant length (Brennan et al., 2008). Their equations are listed below.

$$\text{MutualAttention} = \frac{1}{T} \sum_{t=1}^{T} \mathbf{1}\left\{ \arccos\left(\mathbf{g}_i(t) \cdot \mathbf{d}_{i \to j}(t)\right) \leq \theta_{\text{tol}} \ \wedge \ \arccos\left(\mathbf{g}_j(t) \cdot \mathbf{d}_{j \to i}(t)\right) \leq \theta_{\text{tol}} \right\}$$

$$\text{SharedGaze} = \frac{1}{T} \sum_{t=1}^{T} \mathbf{1}\left\{ \left\| \tfrac{1}{P} \sum_{i=1}^{P} \mathbf{g}_i(t) \right\| \geq \tau \right\}$$

$$\text{AU\_sync} = \frac{2}{P(P-1)} \sum_{i<j} \max_{|\ell| \leq L} \ \text{corr}(M_{:,i}, \ \text{shift}(M_{:,j}, \ell))$$

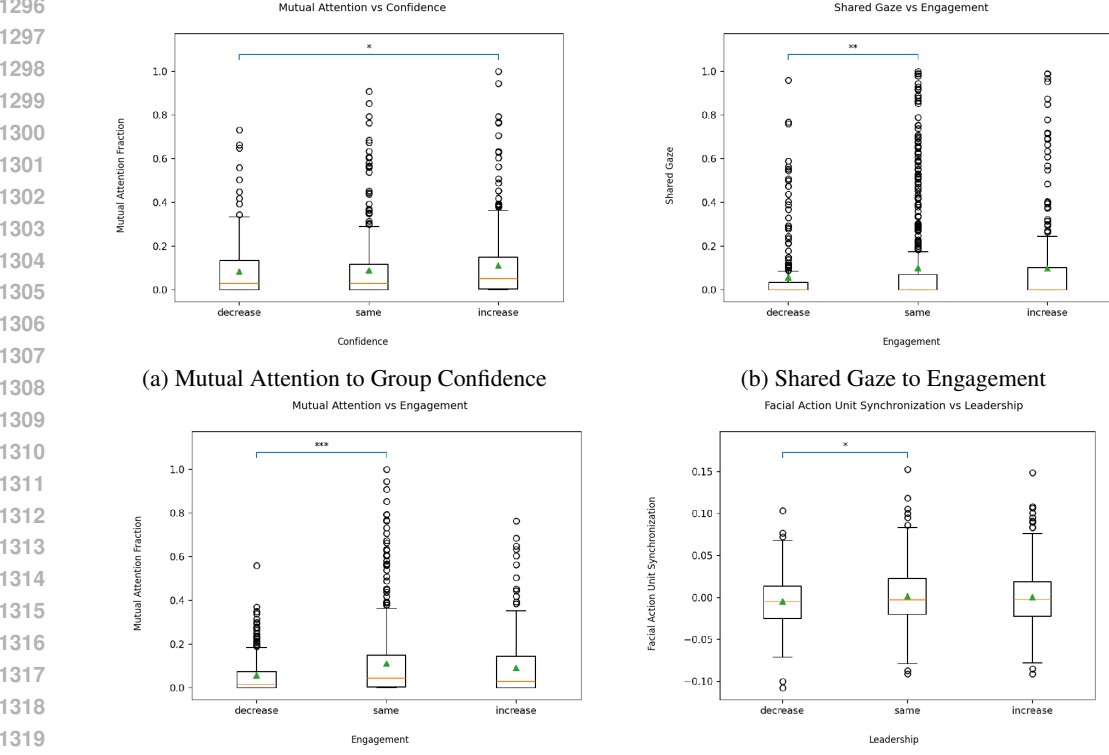

(a) Mutual Attention to Group Confidence      (b) Shared Gaze to Engagement

(c) Mutual Attention to Engagement      (d) Facial Action Unit Synchronization to Leadership

Figure 11: Behavioral Patterns to Group States

### A.6 CLASSIFICATION HEADS

SyntalNet employs two classification heads (individual- and group-level) to classify corresponding constructs of cohesion, both following a shared-trunk, per-head adapter, and a lightweight classifier design. Let $\mathcal{H}$ be the set of heads and let $\tau : \mathcal{H} \to \{\text{ind}, \text{grp}\}$ label each head by its type (individual or group). Each head $h \in \mathcal{H}$ predicts over a task-specific label set $\mathcal{Y}_h$ with $K_h = |\mathcal{Y}_h|$ classes.

**Trunk.** For an individual-level head $h_{\text{ind}}$, the trunk projects a fused representation $\mathbf{Z} \in \mathbb{R}^D$ into a task-shared latent space using a shallow MLP with a learned per-channel residual gate $\alpha \in \mathbb{R}^D$:

$$\mathbf{u}_{\text{ind}} = \mathbf{Z} + \alpha \odot MLP(\mathbf{Z}).$$

For a group-level head $h_{\text{grp}}$, given features $\{\mathbf{Z}_i\}_{i=1}^P$ for $P$ individuals, we use a DeepSets-style trunk $(\phi, \psi)$:

$$\mathbf{s} = \frac{1}{P} \sum_{i=1}^{P} \phi(\mathbf{Z}_i), \qquad \mathbf{u}_{\text{grp}} = \psi(\mathbf{s}),$$

where $\phi$ is an MLP applied per person and $\psi$ is a gated MLP mapping the pooled statistic to $\mathbb{R}^D$.

**Adapter.** Each head applies a lightweight adapter $a_h : \mathbb{R}^D \to \mathbb{R}^D$. We employ a LoRA–FiLM adapter, which modulates features via learned FiLM parameters $(\boldsymbol{\gamma}_h, \boldsymbol{\beta}_h) \in \mathbb{R}^D$ and a low-rank residual update:

$$\tilde{\mathbf{u}}_h = \mathbf{u}_h + \mathbf{u}_h \odot \boldsymbol{\gamma}_h + \boldsymbol{\beta}_h, \quad \mathbf{z}_h = \tilde{\mathbf{u}}_h + \alpha_h \mathbf{B}_h \, \sigma(\mathbf{A}_h \tilde{\mathbf{u}}_h),$$

with $\mathbf{A}_h \in \mathbb{R}^{r_h \times D}, \mathbf{B}_h \in \mathbb{R}^{D \times r_h}$, rank $r_h$, nonlinearity $\sigma$, and scaling factor $\alpha_h > 0$.

**Classifier.** Each head uses a cosine classifier fused with EMA prototypes. With normalized features $\widehat{\mathbf{z}}_h$, class weights $\mathbf{W}_h$, and prototypes $\mathbf{P}_h$, the per-class logit is

$$\ell_{h,k} = (1 - \lambda_{h,k} w) \, s_{\text{param}} \langle \widehat{\mathbf{z}}_h, \widehat{\mathbf{w}}_{h,k} \rangle + (\lambda_{h,k} w) \, s_{\text{proto}} \langle \widehat{\mathbf{z}}_h, \widehat{\mathbf{p}}_{h,k} \rangle,$$

where $s_{\text{param}}, s_{\text{proto}} > 0$ are learned scales, $\lambda_{h,k} \in (0,1)$ are per-class fusion weights, and $w = \frac{1}{2}(1 - \cos(\pi p))$ with $p$ increasing linearly over the first $E$ epochs. Early in training, $w$ is small so logits rely on parametric weights; later, prototype contributions increase.

Prototypes are updated by EMA:

$$\mathbf{p}_{h,k} \leftarrow \text{norm}(m\mathbf{p}_{h,k} + (1-m)\boldsymbol{\mu}_{h,k}),$$

with momentum $m$ and minibatch mean $\boldsymbol{\mu}_{h,k}$. Unseen classes are masked using prototype counts. Finally, a learned temperature $T_h \in [1,2]$ calibrates logits via $\mathbf{y}_h = \text{softmax}(\boldsymbol{\ell}_h/T_h)$.

### A.7 TRAINING DETAILS

#### A.7.1 TRAINING OBJECTIVE

Since the data exhibits a heavy long-tail distribution with strong class imbalance, we employ a *class-balanced focal loss* (Cui et al., 2019) to mitigate domination by frequent classes and improve learning on minority classes. Class-balancing reweighs the contribution of each class according to its effective number of samples, while the focal term down-weights well-classified examples so that training focuses on harder instances. Formally, given logits $\ell \in \mathbb{R}^K$ and target label $y \in \{1, \ldots, K\}$:

$$\mathcal{L}_{\text{CB-Focal}} = \alpha_y (1 - p_y)^\gamma \big( -\log p_y \big), \quad p_y = \frac{e^{\ell_y}}{\sum_{k=1}^K e^{\ell_k}},$$

where $\alpha_y$ is the class-balanced weight derived from the effective number of samples, and $\gamma > 0$ is the focusing parameter. This choice of loss improves robustness under severe imbalance and encourages the model to learn more discriminative features for rare classes (Cui et al., 2019).

### A.8 VLM BASELINE

Listing 1: VLM prompt used for social scene change estimation.

```
1  You will compare TWO consecutive 19s clips from the SAME group.
2  Use BOTH transcript and images for EACH clip.
3  Assume LEFT/MIDDLE/RIGHT by visual position in the images. If fewer than 3 are visible,
4  infer based on available evidence; if uncertain, choose the closest category.
5
6  Return ONLY a JSON object with CHANGES (decrease, same, increase) for:
7  - synchronized_efforts_change (1-5 scale, behind the scenes): are individual efforts
   better coordinated in B vs A?
8  - team_confidence_change (1-5 scale, behind the scenes): is the team more confident in B
   vs A?
9  - interaction_phase_change: compare positions on this ordered scale: [phase_order list
   here]. Later in the sequence = increase; earlier = decrease.
10 - engagement_change: per LEFT/MIDDLE/RIGHT (0-2 scale, behind the scenes: 0=minimal,
   1=moderate, 2=high): did engagement rise/fall/same in B vs A?
11 - social_roles_change: per LEFT/MIDDLE/RIGHT (0=follower, 1=mixed, 2=leader): did role
   tendency rise/fall/same in B vs A?
12 - Add a brief "rationale".
13 Output valid JSON ONLY, no extra keys.
```

(a) First Frame                    (b) Second Frame                    (c) Third Frame

Figure 12: Example Frames for VLM Input. GPT provided reasoning for this specific pair: "In the later clip, the group aligns on values ("close enough... fifty, boom") with smiles; actions proceed smoothly. Participant on the left continues leading hands-on work; middle remains mostly observing; right speaks up to drive a decision, showing higher engagement and sligt leadership. Overall coordination and confidence rise, moving the team further into performing/wrapping up.

