# OpenReview forum: "Reading the Room: Learning Group States Beyond Pooled Individual Signals"
_ICLR.cc/2026/Conference — Submitted to ICLR 2026_

### Official Review · Reviewer_BpXd · 2025-10-28

**Soundness:** 2
**Presentation:** 1
**Contribution:** 1
**Rating:** 2
**Confidence:** 5

**Summary:**

The paper addresses the challenge of modeling group dynamics, where collective states emerge from interdependent interactions, making standard aggregation of individual features insufficient. The authors formalize this as a prediction problem for five constructs: group synchrony, group confidence, interaction phase, individual engagement, and leadership. To tackle this, they propose SyntalNet, a neural architecture with three main components:
a) Mask-aware encoders to handle missing modalities,
b) Permutation-equivariant cross-participant context gating (SoSE-X) to model interdependencies,
c) Low-rank cross-modal fusion (GLRX) to efficiently capture interactions across modalities.
They introduce Syntality, a dataset derived from triadic interactions with temporally smoothed crowd annotations, formulated as directional trend prediction (increase/stable/decrease). Using 10-fold Leave-One-Group-Out cross-validation, SyntalNet outperforms baselines (e.g., Temporal CNN) across all constructs and evaluation metrics. Notably, pooled individual features perform no better than constant predictors, underscoring the necessity of explicitly modeling cross-participant interactions.

**Strengths:**

- They introduce Syntality, derived from triadic interactions with temporally-smoothed annotations for five constructs.
- They report that SyntalNet outperforms baselines, including Temporal CNN, across several metrics.

**Weaknesses:**

1)	The paper overlooks several important earlier works that are essential for connecting established practices with this paper. For instance, it should reference the foundational leadership studies by Daniel Gatica-Perez, as well as the cohesion-related works by Giovanna Varni. Relevant contributions by Maja Pantic and Alessandro Vinciarelli also cover aspects closely related to this study. The absence of such background literature leaves many sentences insufficiently supported by citations.

2)	The Related Work section focuses primarily on affective computing (e.g., stress, trust, emotions, measures of work load (tireness, stress), arousal…), where the analyzed signals are typically much shorter than those encountered in social signal processing phenomena. Many of the papers cited in this section appear to be only loosely related or even irrelevant to the current study. To better align this section with the paper’s scope—particularly regarding group dynamics—the authors could benefit from incorporating insights from a recent survey paper such as: [A] Beyan, Cigdem, Alessandro Vinciarelli, and Alessio Del Bue. "Co-located human–human interaction analysis using nonverbal cues: A survey." ACM Computing Surveys 56.5 (2023): 1-41.

3)	While the proposed framework is well-structured and integrates several established components, I find limited technical novelty in the design described. The key elements: mask-aware encoders, Squeeze-and-Excitation style modules for participant interactions, low-rank cross-modal fusion, and DeepSets compatible prediction heads are all based on existing and well-studied concepts. The paper appears to combine these techniques in a coherent manner rather than introducing a fundamentally new mechanism or theoretical contribution.

4)	The evaluation setup appears quite limited in scope. The experiments are conducted on the Syntality dataset, which includes only 10 triads (approximately 170 minutes of data). This is a very small sample size for drawing strong or generalizable conclusions, especially for a method claiming to model complex group-level or social dynamics. Moreover, it is unclear why existing, larger datasets (see [A]) many of which cover similar social or multimodal interaction scenarios, were not used or discussed. The authors should justify this choice and explain how such a limited dataset can meaningfully support the claims of the study.

5)	The listed contributions are clearly structured, but they appear to reflect a combination of existing methodological components rather than introducing a fundamentally new formulation or architectural mechanism. The formulation point restates well-known challenges in modeling group interactions, and the architectural modules seem to adapt established concepts (e.g., SE blocks, low-rank fusion, DeepSets) with minor variations. The empirical evidence, although positive, is based on a very limited dataset, making it challenging to assess the approach's generality. Overall, the contributions section would benefit from clarifying what is genuinely novel, conceptually, methodologically, or empirically, beyond the integration of existing techniques.

6)	The text contains a broken reference (“see ??”), which prevents readers from accessing important implementation details. Further, I am not able to spot implementation details; the given ones belong to the dataset's cues.

7)	The claim regarding dataset availability is inaccurate. The paper states that there is “no public dataset of collaborative problem solving,” which is not correct. Several publicly available datasets exist that capture collaborative or group problem-solving scenarios. The authors should revise this statement to accurately reflect prior resources and position their dataset accordingly.

8)	The description of inter-rater agreement is unclear. It is not specified which metric was used to assess agreement, whether it is percent agreement, Cohen’s kappa, Fleiss’ kappa, or another statistical measure. Additionally, the meaning of “66.7% threshold for strong labels” and how it relates to the reported percentages for the five classes is ambiguous. Clarification is needed to understand the reliability of the annotations.

9)	The reported weak correlation between individual and group states, alongside strong links between group states and synchronized behaviors (e.g., mutual attention, shared gaze), suggests that modeling individual features alone is insufficient for predicting group-level outcomes. This emphasizes the need for methods that explicitly capture inter-participant interactions and temporal synchronization. Given this, the paper should clarify how the proposed model effectively leverages these synchrony signals and discuss potential limitations due to the small dataset size.

10)	The baseline comparisons are somewhat limited and raise concerns about fairness. Logistic Regression uses hand-crafted group features, while SyntalNet leverages learned multimodal representations, potentially biasing results. It is also unclear whether the LSTM, Temporal CNN, and VLM baselines were fully optimized for the small dataset (no implementation details supplied), which could exaggerate SyntalNet’s performance advantage. The authors should justify baseline selection and tuning, and consider including alternatives like DeepSets or cross-modal attention models that are more directly relevant to group-level modeling.

11)	The paper presents results in a single table without any ablation studies. While some components (e.g., SoSE-X, GLRX, DeepSets heads) may not be directly removable, the authors could still assess their contribution by replacing them with reasonable alternatives or simpler versions. Ablation experiments of this kind are necessary to understand which parts of SyntalNet drive the reported improvements and to evaluate the necessity of each module.

**Questions:**

Please see the weaknesses. Importantly:

Could you clarify which inter-rater agreement metric was used and how the “66.7% threshold for strong labels” relates to the reported class percentages?

Can you explain the intended contribution of each SyntalNet component (SoSE-X, GLRX, DeepSets heads) and why these design choices were made, given that no ablation studies are included?

How does the model leverage synchronized behaviors to predict group states, and how does this relate to the weak correlation between individual and group features?

---

> ### Author Response · Authors · 2025-11-22
> **Official Comment by Authors - 1**
>
> We are thankful for the reviewer **BpXd**'s detailed and constructive feedback, especially related to the challenges of modeling group dynamics and the necessity of explicitly capturing cross-participant interactions. We were glad to see that you acknowledge the importance of explicitly modeling such interactions. Several of the 11 weaknesses center around a few common themes. For clarity and to avoid repetition, we group them into:
>
> - **Theme A:** Positioning, prior work, and dataset context
> - **Theme B:** Novelty, contributions, and modeling
> - **Theme C:** Experimental details and empirical validation
>
> We address each theme in turn below, and in doing so respond explicitly to all of your numbered concerns.
>
> ---

---

> ### Author Response · Authors · 2025-11-22
> **Official Comment by Authors - 2**
>
> ### **Theme A: Positioning, prior work, and dataset context**
>
> ---
>
> **Literature coverage & related work scope (comments 1 and 2)**
>
> We appreciate the reviewer's recommendation on the literature coverage, specially toward social signal processing and group interaction analysis. We have substantially expanded both the Section 1 (Introduction) and Section 2 (Related Work) to better connect Syntality to prior work on group state sensing and will share the details in the revised manuscript.
>
> Specifically, we included early multimodal work on leadership and dominance in small groups, including Jayagopi et al. 2009, Sanchez-Cortes et al. 2010/2013, and Hung et al. 2010, which directly reflect the contributions of Daniel Gatica-Perez and Giovanna Varni on group leadership and cohesion. We additionally discuss the GAME-ON dataset and the follow-up modeling work by Maman et al. 2020/2021 on group cohesion and engagement. To better situate our work within social signal processing, we also incorporate survey and conceptual contributions by Maja Pantic and Alessandro Vinciarelli, including Vinciarelli et al. 2008, as well as the recent survey on co-located human–human interaction by Beyan et al. 2023 recommended by the reviewer.
>
> We believe these editions enhance the quality of the manuscript by clarifying how our problem formulation, dataset, and architecture build directly on, and extend, group-level social dynamics modeling line of work.
>
> ---
>
> **Dataset positioning (comment 7)**
>
> We agree that our original wording about dataset availability was potentially misleading. We did not intend to claim that *no* public dataset of collaborative problem solving exists, but rather that, to our knowledge, no existing corpus is designed for studying **non-additive compositionality** in triadic collaboration with the combination of signals and labels we require.
>
> We have reviewed all related datasets discussed in the survey of Beyan et al. (2023). Most corpora are either dyadic or competitive (e.g., social games, debates), provide only static/global or sparse event labels, or annotate group-level *or* individual-level constructs but not *pairs* of individual+group trends. In contrast, Syntality extends The Weights Task Dataset (Khebour et al.\ (2024)) to triadic, physically grounded collaboration with (i) participant-indexed dense multimodal streams, (ii) temporally smoothed *directional trend* labels (increase/stable/decrease) for five constructs, and (iii) dense supervision of paired individuals+group trends with large-scale crowd annotation. These are needed to test whether group states can be modeled as a simple additive function of individual states.
>
> In the revised manuscript, we explicitly position Syntality alongside datasets including Idiap Wolf, Twenty-Question Game, ELEA, TeamSense, Panoptic, Resistance Game, UDIVA, and GAME-ON, and make the conclusion stated above. We acknowledge that GAME-ON is the closest in spirit, but, to our knowledge, it does not provide the same combination of triadic physical collaboration, participant-indexed streams at all times, and joint directional trend labels for both individual and group constructs. We have revised Section 3 (Dataset And Annotation), Section 2 (Related Work), and the appendix to correct our claim and clarify this positioning.
>
> ---

---

> ### Author Response · Authors · 2025-11-22
> **Official Comment by Authors - 3**
>
> **Dataset size and justification (comment 4)**
>
> We acknowledge that Syntality contains only 10 triads, which limits the diversity of teams and contexts. Therefore, we don't claim universal generalization across all forms of group dynamics. But rather we focus on empirically testing **the non-additive compositionality conjecture** in a controlled triadic collaborative setting. We assess whether explicitly modeling cross-participant interactions and synchrony improves prediction in this regime.
>
> For this purpose, we argue that Syntality is both *necessary* and *sufficient*:
> - It is *necessary* because the signal we are interested in requires (i) at least triadic interaction, where higher-order composition effects are meaningful; (ii) participant-indexed multimodal streams that separate individuals from their synchrony; and (iii) temporally dense, paired individual+group trend labels for such emergent states we study. Existing larger datasets do not jointly offer these properties. Therefore, using them to test non-additive compositionality would require collecting new dense annotations for individual and group trends, which means constructing a new dataset. This is beyond the scope of this paper. Syntality is designed specifically to contain the signals needed to empirically probe this conjecture.
> - It is *sufficient* because, while the number of groups is small, the supervision is temporally rich. Using a sliding-window formulation, we obtain 3,360 labeled clips of group multimodal streams. Each clip comprises three participant streams and five paired individual+group trend labels, with 1,128 crowd workers contributing multi-rater annotations. Empirically, SyntalNet converges reliably on Syntality Dataset, outperforms pooled-feature baselines, and supports out-of-domain (OOD) transfer (detailed under Theme~C). These results indicate that the dataset size is adequate for our specific objective and provides enough signal for learning meaningful representations. Thanks to Syntality, we demonstrated that additive aggregation of individual features is insufficient, and that architectures explicitly modeling cross-participant interactions and synchrony yield systematic gains in this triadic collaborative task.
>
> ---

---

> ### Author Response · Authors · 2025-11-22
> **Official Comment by Authors - 4**
>
> ### **Theme B: Novelty, contributions, and modeling**
>
> ---
>
> **Novelty and contributions (comments 3 and 5)**
>
> We appreciate the reviewer's concern about technical novelty. The main novelty of our work lies in the **non-additive compositionality formulation** of group-state prediction and the **Syntality Dataset** that makes the hypothesis observable. We empirically show that pooled individual signals yield only modest improvements over the baselines and remain far below models that explicitly model cross-participant interactions (see Theme C for details). This exposes a structural mismatch between common modeling assumptions and observed dynamics of group states. The observation that group constructs are fundamentally non-additive, motivates the design requirements, including effective cross-participant fusion and robust multichannel, multimodal data fusion. Therefore, our other novelty is the non-trivial architecture of **SyntalNet** that is derived from the above hypothesis and observations in the Syntality Dataset. We will revise the Contributions section to emphasize this formulation more explicitly.
>
> SyntalNet instantiates this formulation with three modules justified by what's needed to capture non-linear composition under small-data constraints.
>
> The *Social Squeeze-and-Excitation-based Cross-participant Fusion (SoSE-X)* is not a standard SE block. Standard SE re-weights channels based on self-statistics. SoSE-X computes a permutation-equivariant "context" vector and conditions each participant's stream representation on this cross-participant signal via a residual gate. The learnable scalar residual coefficient allows the model to adaptively control cross-participant coupling strength. This yields an attention-like cross-participant fusion with far fewer parameters and lower data demands than self-attention.
>
> The *Branch-Separable Channel Mixer (GSX)* performs mask-aware fusion of substreams within a modality (e.g., facial landmarks, gaze, body pose, etc.). It handles intra-modality heterogeneity, allows exploiting complementary coverage of substreams, and remains robust to partial visibility and temporal misalignment.
>
> The *Gated Low-Rank Cross-modal Fusion (GLR-X)* extends low-rank bilinear fusion to multiple modalities by pairwise interactions between all active modalities, without the complexity of full cross-modal attention. It decouples unary vs. pairwise pathway with a learned mixing weight. It has an allocation gate that adaptively prioritizes modalties based on their quality/expressiveness. Ablations as detailed under Theme C show that these structural choices yield consistent gains over other instantiations.
>
> ---
>
> **Use of synchrony and cross-participant signals (comment 9)**
>
> We thank the reviewer for highlighting this point. Our model does leverage synchrony signals, but not as explicit hand-crafted features. Instead, synchrony emerges implicitly through the multimodal streams and SoSE-X cross-participant fusion.
>
> SoSE-X enables participants' representations to interact before pooling. The module computes a masked mean of other participants' SE-gated features, applies depthwise convolution for temporal filtering and synchrony to capture not perfectly aligned behaviors, and injects this as a residual into each participant's stream. This allows the model to learn relational patterns, for example if participant A's gaze aligns with participant B's head position (mutual attention), or if their pitch/energy trajectories co-vary (prosodic synchrony). These patterns cannot be captured by pooling individual features independently, but the network creates internal feature maps that activate specifically when an individual aligns with or diverges from the group.
>
> Furthermore, GSX fuses multiple substreams within each modality using masked Generalized Mean pooling and lightweight ConvNeXt-v2–style blocks. This helps retain synchrony patterns that are distributed across channels and reduces the impact of temporal misalignment between substreams. GLRX then applies low-rank pairwise interactions across modalities, which enables the model to highlight joint evidence (e.g., mutual gaze and elevated prosodic energy) instead of treating modalities independently. Because the dataset is limited in size, all three modules are designed to be parameter-efficient.
>
> ---

---

> ### Author Response · Authors · 2025-11-22
> **Official Comment by Authors - 5**
>
> ### **Theme C: Experimental details and empirical validation**
>
> ---
>
> **Baselines and ablations (comments 10 and 11)**
>
> The reviewer correctly notes that Logistic Regression uses hand-crafted group features. Logistic Regression is deliberately simple to act as the aggregation fallacy baseline, which treats the group as a bag of individuals without explicit relational modeling. The poor performance confirms our hypothesis that simple pooling is insufficient.
>
> However, our other baselines are not toy models. Mask-aware Temporal CNNs and BiLSTMs are widely used and competitive sequence models in human activity recognition and affective computing. In our setup, they share the same participant-indexed multimodal features input, roughly similar classifier structure, and training protocol as SyntalNet, follow a similar classifier structure, and are trained with the same optimizer, schedule, and regularization. We also tune their hidden size, depth, and dropout on validation folds. Therefore, they are roughly capacity-matched to SyntalNet. Conceptually, they represent strong temporal models that still rely on additive group composition.
>
> In addition, as a part of an ablation test, we include a DeepSets-style variant of SyntalNet, which aggregates participant embeddings with permutation-invariant pooling but removes explicit cross-participant coupling. Together with the GPT-5 VLM zero-shot evaluation of behavioral prediction and the ablations, these baselines cover the main modeling assumptions we seek to test.
>
> Given the size of the dataset and the structured feature space, training large multimodal transformers would be impractical and highly prone to overfitting. We therefore believe this baseline suite is appropriate for the current dataset and task, and that the observed performance gaps reflect genuine benefits of SyntalNet's compositional design rather than overly weak baselines.
>
> We will add the implementation details of all models in the revised manuscript, and will release full training scripts (if we get accepted).
>
> Regarding the need for ablation studies, during the rebuttal period, we managed to conduct a number of various experiments and will add the details of the four sets of new experiments in the revised manuscript.
>
> 1. *Cross-participant fusion.* We ablate both the cross-participant fusion (SoSE-X) and the group classification head. The group head normally takes a permutation-invariant pooled representation of all participants and outputs group-level predictions. When we ablate it, we instead obtain group labels using an individual head across all participants, which enforces a purely additive bag-of-individuals assumption. In the ablation, we toggle SoSE-X and the group head, which yields four variants: neither, group head only (DeepSets-style pooling), SoSE-X only, and the full model. As expected, removing cross-participant components substantially reduces group-level performance. Any form of cross-participant aggregation improves over "neither", and variants with SoSE-X show the largest gains. The full model performs best on all group constructs.
>
> 2. *Intra-modality fusion (GSX).* We compare GSX to Concat+MLP and Mean Pooling under three perturbations: temporal misalignment jitter, relative Gaussian feature noise, and temporal band masking. GSX consistently achieves higher performance metrics under various degrees of perturbations, demonstrating that its design brings tangible robustness benefits.
>
> 3. *Cross-modal fusion (GLRX).* We replace GLRX with Concat+MLP, Mean Pooling, a gated-sum only version, and a pairwise-only version, and stress-test under modality dropout, modality noise, and modality shuffling. GLRX maintains stronger performance under missing and contradictory modalities, indicating that combining availability-aware allocation with low-rank pairwise terms is beneficial.
>
> 4. *Out-of-domain transfer.* We additionally report the zero-shot transfer performance and adaptation with leave-one-group-out (LOGO) experiments. As in prior HAR/BCI/affective work, LOGO performance is substantially lower than random splits, which reflects a genuinely hard cross-subject problem. We then show that fine-tuning SyntalNet on a small fraction of the left-out group's data with few epochs yields meaningful gains and approaches within-subject baselines. Also, to specifically assess the transferability of the learned representation, we freeze the entire encoder and only finetune the classification head on the left-out group. This limits the model's capacity to memorize target-specific details. Therefore, the performance gains come from slightly adjusting the decision boundary and not feature space itself. These results suggest that the learned representations are reusable and that SyntalNet supports efficient OOD adaptation, even though the OOD gap is not fully closed.
>
> ---

---

> ### Author Response · Authors · 2025-11-22
> **Official Comment by Authors - 6**
>
> **Inter-rater agreement (comment 8)**
>
> We clarify the annotation scheme as follows. For each clip and construct, we compute simple percent agreement. If at least 66.7% of annotators (e.g., $\geq 2$ of 3) select the same trend category, we mark the label as "strong"; otherwise it is treated as "weak" and only used through our temporal smoothing kernel. The reported percentages (80.37%, 79.20%, 76.71%, 91.44%, 89.57%) are the proportions of clips with strong labels for each construct.
>
> ---
>
> **Implementation details and broken reference (comment 6)**
>
> We apologize for the broken reference, which should point to the appendix containing our signal-processing pipeline. We have fixed this citation. To address missing implementation details, we will add a dedicated appendix section listing all SyntalNet hyperparameters, the exact training schedule and regularization, and corresponding details for all baselines.
>
> ---

---

> ### Author Response · Authors · 2025-11-22
> **Official Comment by Authors - 7**
>
> ### **Response to Questions**
>
> > ***Could you clarify which inter-rater agreement metric was used and how the "66.7% threshold for strong labels" relates to the reported class percentages?***
>
>   We used percent agreement at the clip level. For each construct, if at least 66.7% of annotators (e.g., $\geq 2$ of 3) chose the same trend label, that clip was marked as having a strong label, otherwise it was treated as weak. The reported percentages are the fractions of clips with strong labels per construct. In the revision, we also additionally report a standard reliability coefficient, in Fleiss's $\kappa$, which are 0.286, 0.284, 0.348, 0.439, and 0.468 for the 5 classes. For the strong labels, their Fleiss's $\kappa$ are 0.502, 0.357, 0.676, 0.914, and 0.903, indicating very strong alignments.
>
> > ***Can you explain the intended contribution of each SyntalNet component (SoSE-X, GLRX, DeepSets heads) and why these design choices were made, given that no ablation studies are included?***
>
>   SoSE-X provides permutation-equivariant cross-participant fusion. It injects a residual summary of the other participants' gated features into each person's stream, which acts as a lightweight, data-efficient alternative to self-attention. It directly targets non-additive interpersonal composition. GLRX performs cross-modal fusion by combining quality/expressiveness allocation with low-rank pairwise interactions, so the model can represent both unary modality contributions and pairwise synergies under missing/noisy modalities. The DeepSets-style heads aggregate participant embeddings into a group representation in a permutation-invariant way once cross-participant interactions have been encoded. These choices follow from the compositionality hypothesis and small-data constraints. The new ablations as discussed in Theme C show that removing or simplifying these modules systematically degrades performance and robustness.
>
> > ***How does the model leverage synchronized behaviors to predict group states, and how does this relate to the weak correlation between individual and group features?***
>
>   Syntality provides synchrony-relevant cues as per-participant streams. SoSE-X implicitly leverages these by letting each participant's representation depend on a temporally filtered summary of the others' features. Therefore, synchrony (and divergence) patterns are encoded before pooling. GSX preserves synchrony distributed across substreams within a modality, and GLRX upweights joint cross-modal evidence. In contrast, pooled individual features ignore these interactions and therefore show weak correlation with group labels, thus ignore the non-additive compositionality effect we emphasis on and model using SyntalNet.
>
> ---

---

> ### Author Response · Authors · 2025-11-22
> **Official Comment by Authors - 8**
>
> ### **Closing Remarks and Next Steps**
>
> We appreciate your thorough feedback and the opportunity to improve the paper. With the added experiments, clarifications, and literature connections, we have aimed to fully respond to your specific concerns. Do you feel that these updates adequately address your feedback, or are there further changes you would suggest?

---

### Official Review · Reviewer_dnpp · 2025-10-31

**Soundness:** 2
**Presentation:** 2
**Contribution:** 2
**Rating:** 4
**Confidence:** 3

**Summary:**

This paper introduces a machine learning model to infer five constructs related to human interaction: two individual-level constructs and three group-level constructs. The authors aim to demonstrate that group-level constructs cannot be inferred from individual observations alone, but rather through collective states that emerge from interactions between partners. In addition to their model, they manually annotate a publicly available dataset to create grounded labels for model training. They compare their results against various baseline models and find that their proposed model outperforms them.

**Strengths:**

The proposed architecture is able to encompass data from different modalities, each with its own time scale, dimensions, and preprocessing steps. There are also distinct architectural components designed to fuse information across different dimensions, both across individuals and across modalities, which may serve as inspiration for future work on group interaction. Additionally, the annotated dataset, created by multiple annotators through an online platform, is a valuable contribution of this paper and could benefit future work if made publicly available.

**Weaknesses:**

There are no ablation studies included in the paper. The authors claim that their findings show emergent team states cannot be reduced to pooled individual signals, but this is not demonstrated, as all reported results are based on their full-fledged model. For this claim to be supported, I would expect to see lower group-level prediction performance when cross-module components are removed. The architecture is also quite complex, making it difficult to assess the individual contribution of each component without ablation. Additionally, while the abstract mentions using 10-fold Leave-One-Group-Out cross-validation, this detail is missing from the evaluation section. The authors also omit reporting variance in their results, which makes it hard to assess statistical significance. Finally, the paper does not cite relevant prior work on multimodal dynamic systems for group interaction and group-level construct inference, such as [1] and [2].

[1] Paulo Soares, Adarsh Pyarelal, Meghavarshini Krishnaswamy, Emily Butler, and Kobus Barnard. 2024. Probabilistic modeling of interpersonal coordination processes. In Proceedings of the 41st International Conference on Machine Learning (ICML'24), Vol. 235. JMLR.org, Article 1867, 45906–45921.

[2] Moulder, R. G., Duran, N. D., & D'Mello, S. K. (2022). Assessing Multimodal Dynamics in Multi-Party Collaborative Interactions with Multi-Level Vector Autoregression. In ICMI 2022 - Proceedings of the 2022 International Conference on Multimodal Interaction (pp. 615-625). (ACM International Conference Proceeding Series). Association for Computing Machinery. https://doi.org/10.1145/3536221.3556595

**Questions:**

# Questions
1. Why predicting trends on 10-s windows, 3-s stride and 19-s observation? I just want to understand how those numbers were chosen.
2. In L157,  the authors say "...observe 10 clips, and each clip was annotated by at least 3 participants". What's the duration of each clip and how did you guarantee that all clips have some relevant information to be extracted? Or were some clips discarded after labeling?
3. What do the increase, flat and decrease actually mean for the labels? Does increase here mean that label score in clip t < label score in clip t+1?

# Typos
L106. In paralle -> In parallel

L146. Broken link

L189. Distribution of the label 4. -> The distribution of labels is in Figure 4

---

> ### Author Response · Authors · 2025-11-22
> **Official Comment by Authors - 1**
>
> We appreciate Reviewer **dnpp**'s careful reading and constructive remarks, especially the suggestions regarding ablations, evaluation protocol details, and links to prior work on multimodal dynamic systems. We are also encouraged that you found the multimodal design potentially inspiring for future work and recognized the value of the crowdsourced annotations. Below we address the main weaknesses and questions.
>
> ---
>
> ### **Ablations**
>
> We agree that ablations are important, and we conducted several experiments during the rebuttal period and will add the details in the revised manuscript. This should make explicit that our claim about emergent team states not reducing to pooled individual signals is supported.
>
> 1. We ablate both the cross-participant fusion (SoSE-X) and the group classification head. The group head normally takes a permutation-invariant pooled representation of all participants and outputs group-level predictions. When we ablate it, we instead obtain group labels using an individual head across all participants, which enforces a purely additive bag-of-individuals assumption. In the ablation, we toggle SoSE-X and the group head, which yields four variants: neither, group head only (DeepSets-style pooling), SoSE-X only, and the full model. As expected, removing cross-participant components substantially reduces group-level performance. Any form of cross-participant aggregation improves over "neither", and variants with SoSE-X show the largest gains. The full model performs best on all group constructs.
>
> 2. We compare the intra-modality fusion (GSX) against Concat+MLP and Mean Pooling under temporal jitter, feature noise, and temporal band masking. GSX consistently achieves higher performance metric and degrades more gracefully.
>
> 3. We replace the cross-modal fusion (GLRX) with Concat+MLP, Mean Pooling, a gated-sum-only version, and a pairwise-only version, and stress-test under modality dropout, noise, and shuffling. GLRX maintains the strongest performance, especially when modalities are missing or contradictory.
>
> ---
>
> ### **Evaluation Protocol and Statistics**
>
> We describe the evaluation protocol more clearly in the revised manuscript. We have used 10-fold LOGO as well as K-Fold validation with k=5. More specifically, we used the K-Fold setup for all main results. We have updated the results table to report the mean and standard deviation across the folds for all models. In addition, we include LOGO-based adaptation experiments where we finetune SyntalNet on small fractions of the held-out group's data, as well as experiments where we freeze the encoder and only adapt the classifier heads. Preliminary results show that under LOGO, SyntalNet exhibits a clear zero-shot performance gap due to covariate shift but recovers with modest adaptation, showing Out of Domain (OOD) transferability.
>
> ---
>
> ### **Prior work**
>
> We appreciate the pointers to Soares et al.\ (ICML'24) and Moulder et al.\ (ICMI'22). We have now cited and briefly discussed both and situated SyntalNet alongside these dynamic modeling approaches.
>
> ---

---

> ### Author Response · Authors · 2025-11-22
> **Official Comment by Authors - 2**
>
> ### **Response to Questions**
>
> > ***Why predicting trends on 10-s windows, 3-s stride and 19-s observation? I just want to understand how those numbers were chosen.***
>
>   We chose 10-sec windows to follow the "thin-slice" literature on small-group interaction and cohesion, where human raters reliably infer group-level constructs from segments in the 8–21 sec range. We utilize the "interval coding"-like segmentation common in Panoptic-based cohesion studies with a fixed window. A 10-sec slice is long enough to capture turn-taking, short sequences of mutual gaze, and brief verbal exchanges, while still being short enough that annotators can treat each clip as approximately homogeneous. We use a 3 s stride to create overlapping windows, which yields smooth trend trajectories and sufficient training samples without excessive redundancy. The 19-sec observation horizon matches the temporal context we expose to the model when judging the change between two partially overlapping 10-sec clips. We clarify these details and provide references in the revised paper.
>
> > ***In L157, the authors say "...observe 10 clips, and each clip was annotated by at least 3 participants". What's the duration of each clip and how did you guarantee that all clips have some relevant information to be extracted? Or were some clips discarded after labeling?***
>
>   Each clip presented to crowd annotators has a duration of 10 sec. Clips are sampled uniformly along the interaction, and we do not filter out quiet segments, because lack of overt behavior can itself convey a social state. Annotators answer 5 questions on 3- or 5-point Likert scales and can choose the midpoint when no salient cue is perceived, which is important for modeling stable or low-activity periods.
>
> > ***What do the increase, flat and decrease actually mean for the labels? Does increase here mean that label score in clip t < label score in clip t+1?***
>
>   For each construct and clip, categorical crowd responses (with agreement scores) are converted into soft label distributions and convolved over time with a confidence-weighted Gaussian kernel to obtain temporally smoothed posteriors. We then compare the mean of these smoothed posteriors for pairs of consecutive 10-s clips. If the value in clip t+1 is higher than in clip t beyond a small tolerance band, we label the trend as increase; if it is lower, as decrease; and if it stays within the band, as no-change. Therefore, "increase" does indeed correspond to a higher underlying construct level in the following clip after smoothing.
>
> ---

---

> ### Author Response · Authors · 2025-11-22
> **Official Comment by Authors - 3**
>
> ### **Closing Remarks and Next Steps**
>
> Thank you once more for the time and thought you invested in reviewing our work. We believe the changes and new analyses outlined above resolve the issues you raised. Do you see any aspects that still need additional evidence or refinement to make the work more convincing?

---

> > ### Comment · Reviewer_dnpp · 2025-11-24
> >
> > Thank you for answering my questions and concerns, improving the related work, and extending the evaluation with ablations and knowledge transfer. My concerns were addressed and I updated my score accordingly.

---

### Official Review · Reviewer_PqZj · 2025-10-31

**Soundness:** 2
**Presentation:** 1
**Contribution:** 2
**Rating:** 4
**Confidence:** 3

**Summary:**

The paper seems to propose a new video dataset capturing a problem-solving task between a group of people to analyze the group synchrony, confidence, interaction phase, and individual engagement and leadership. A baseline is proposed to solve the task and compare against temporal CNN, LSTM, and VLM.

**Strengths:**

The paper proposes a new video dataset capturing a problem-solving task between a group of people, which seems new to me, and many efforts have been put into building the data.

**Weaknesses:**

The paper seems hard to follow for new readers, especially since it is proposing a new dataset and (seemingly) a new task. From my view as a computer vision researcher, the paper does not well formulate the technical problem (not the participant's task). I do not know what the group dynamic is or what the expected visualization of the output is.

It can be fixed if the author can include some better visualization of prediction, annotation, eg, a confusion matrix, or distribute the dataset at the submission time, but unfortunately, it was not. I have a feeling that classification categories are LEFT/MIDDLE/RIGHT associated with participant social roles are just a handful number of classes, and that is limited.

Figure 1 is not informative in showing the features it aggregates or the mathematical operations.

There is no clear comparison with previous datasets to help position the proposed task and method within existing research. From the perspective of established computer vision and temporal modeling tasks, numerous state-of-the-art approaches already exist in time-series estimation and scene-level categorical reasoning (e.g., Scene Graph Generation). In this context, the experiment and method appear limited in novelty and technical innovation.

**Questions:**

See weaknesses.

**Details Of Ethics Concerns:**

Maybe need the consent of the participant's face, voice to be recorded.

---

> ### Author Response · Authors · 2025-11-22
> **Official Comment by Authors - 1**
>
> We thank the reviewer **PqZj** for carefully engaging with the paper and for highlighting both the value of the curated Syntality dataset and several aspects that could be clarified or strengthened, especially regarding task formulation, visualization, and positioning. Below is our response to the concerns.

---

> ### Author Response · Authors · 2025-11-22
> **Official Comment by Authors - 2**
>
> ### **Problem Clarity and Task Formulation**
>
> We apologize that the technical problem was not sufficiently clear to readers outside social signal processing. We will revise Section 3 (Dataset and Annotation) and Section 4 (Problem Setup) to make the task and notation explicit. Here is a concise setting and notation:
>
> Each interaction involves three participants collaborating around a physical "weights" puzzle. We index them by their fixed position in the video frame: **Left**, **Middle**, and **Right** (the left-/center-/right-most person in the camera view). For each participant $p \in \{L,M,R\}$ and clip index $c$, we extract temporally aligned multimodal streams (pose, gaze, facial cues, utterances, speech prosody, etc.) and denote the resulting feature tensor by $\mathbf{x}_p(c)$. We will compute group-level features derived from all three streams.
>
> For each 19 sec clip $c$, we posit latent continuous states for the individual and group constructs. For simplicity of notation, we write the latent representations of continuous states as:
>
> $$
> \mathbf{z}^{h_{ind}}_{p,c} \quad \text{for individual engagement, leadership constructs,}
> $$
>
> $$
> \mathbf{z}^{h_{grp}}_{c} \quad \text{for group synchrony, confidence, and interaction phase constructs,}
> $$
>
> where $h_{ind}$ and $h_{grp}$ represent each of the 2 or 3 individual or group head labels respectively. In practice, we maintain a shared trunk and construct-specific heads for the classifiers of the individual and group level constructs, but we omit that index to keep notation light.
>
> Crowd annotators rate 5 questions (corresponding to these constructs) on 10 sec clips using 3- or 5-point Likert scales. We convert these ratings into temporally smoothed posteriors and derive *trend* labels by comparing pairs of consecutive 10 sec clips with 1 sec overlap. For each construct we then predict one of three classes,
>
> $$
> \text{increase},\ \text{no-change},\ \text{decrease},
> $$
>
> given the corresponding individual state representation
>
> $$\mathbf{z}^{h_{ind}}_{p,c}$$
>
> or group state representation
>
> $$\mathbf{z}^{h_{grp}}_{c}$$
>
> Therefore, the technical problem is: *given the participant-indexed multimodal sequences $\{\mathbf{x}_L,\mathbf{x}_M,\mathbf{x}_R\}$, learn a model that predicts directional trends (increase / no-change / decrease) for five constructs at both the individual and group levels over time.*
>
> A standard **additive** model for group states first encodes each participant independently and then pools these encodings:
>
> $$
> \mathbf{z}^{h_{ind}}_{p,c} = \phi\!\big(\mathbf{x}_p(c)\big),
> \qquad
> \mathbf{z}^{h_{grp}}_{g} = g\!\left(\frac{1}{3}\sum_{p}s^{h_{ind}}_{p,c}\right),
> $$
>
> $$
> \hat{\mathbf{y}}^{\mathrm{ind}}_{p,c} = \mathrm{CLS^{ind}}\big(\mathbf{z}^{h_{ind}}_{p,c}\big),
> \qquad
> \hat{\mathbf{y}}^{\mathrm{grp}}_{c} = \mathrm{CLS^{grp}}\big(\mathbf{z}^{h_{grp}}_{g}\big),
> $$
>
> where:
> - $\phi(\cdot)$ is a per-participant encoder that maps multimodal input to a latent representation $\mathbf{z}^{h_{ind}}_{p,c}$;
> - $g(\cdot)$ is a simple nonlinear function (identity or a small MLP) that maps the pooled individual state representations to a group-level latent state representation $\mathbf{z}^{h_{grp}}_{g}$;
> - $\mathrm{CLS^{ind}}(\cdot)$ and $\mathrm{CLS^{grp}}(\cdot)$ are a classification head that outputs the 3-way trend probabilities for the individual and group construct.
>
> This family of models assumes *additive compositionality*, where the group state is a (possibly nonlinearly transformed) average of independently encoded individual states. Our work instead tests and models a more general, non-additive compositional function that includes explicit cross-participant interactions.

---

> ### Author Response · Authors · 2025-11-22
> **Official Comment by Authors - 3**
>
> In contrast, our central contribution is the *non-additive compositionality conjecture*, where we posit that group constructs in triadic collaboration cannot be reduced to such additive pooling. Instead, SyntalNet learns a more general compositional function
>
> $$
> \hat{\mathbf{y}}^{\mathrm{grp}}_{c}=\mathrm{CLS}^{\mathrm{grp}}\!\Big(\mathbf{F}\big(\mathbf{x}_L(c),\,\mathbf{x}_M(c),\,\mathbf{x}_R(c)\big)\Big)
> $$
>
> where
>
> $$
> F: (x_L(c), x_M(c), x_R(c)) \mapsto h^{grp}_{c}
> $$
>
>
> is a permutation-equivariant mapping that jointly processes the participant-indexed multimodal inputs and produces a group-level representation $\mathbf{h}^{\mathrm{grp}}_{c}$, and $\mathrm{CLS}^{\mathrm{grp}}$ is a group classification head that outputs the 3-way trend probabilities.
>
> Concretely, $\mathbf{F}$ is instantiated by SyntalNet's backbone. SoSE-X performs cross-participant fusion in the encoder so that each participant's embedding depends on the *others*' streams, GSX fuses heterogeneous substreams within each modality in a mask-aware manner, and GLRX introduces low-rank pairwise interactions across modalities. Group predictions are then obtained by applying $\mathrm{CLS}^{\mathrm{grp}}$ to the resulting group representation, while individual trends are predicted via shared head $\mathrm{CLS}^{\mathrm{ind}}$ applied to each participant-specific embeddings. This enables us to model group constructs as a learned non-linear function of all three participants and their interactions, rather than as an average of independently encoded individual states.
>
> ---
>
> ### **Visualizations**
>
> We appreciate the suggestion to improve visualization. We uploaded 3 sample videos of the dataset as supplementary material, one where SyntalNet predicts all constructs correctly and two partial-success cases. Each video will overlay the ground-truth trends and model predictions for individual and group constructs.
>
> We agree that the current Figure 1 is not sufficiently informative. We will revise it in the revision to explicitly depict the input data and data flow, and link each block to its mathematical operation. We also will clearly show the separate individual and group classification heads. We believe this will make the architecture much easier to follow for readers.
>
> ---
>
> ### **Positioning**
>
> We appreciate the concern about positioning. In the revised Related Work we explicitly compare Syntality to existing group-interaction datasets (Idiap Wolf, ELEA, Twenty-Questions, UDIVA, GAME-ON, etc.) and summarize their differences. Most existing datasets either are dyadic or competitive rather than collaborative triads, provide static/global labels (overall cohesion, one group emotion label per clip), or annotate group- *or* individual-level constructs, but not paired individual+group trends with dense temporal resolution. Syntality is, to our knowledge, the first dataset to align participant-indexed multimodal streams with *directional trend* labels for both individual and group constructs, enabling explicit study of how group states emerge from interacting individuals.
>
> Regarding the suggestion to use scene-level categorical reasoning, we see such models as complementary but not directly aligned with our current objective. Scene-graph models typically produce static, discrete relational labels (e.g., "person A looks at person B") for a single frame or short clip, and then aggregate them for a scene-level category. In our setting, the main targets are *temporally evolving, emergent states* such as group confidence and synchrony, expressed over multiple seconds and across multiple modalities. Our goal is not to predict a static scene category but to infer how these constructs increase, stay stable, or decrease over time. Furthermore, given the relatively small number of triads, heavy graph/Transformer models would likely overfit; our architecture is deliberately designed to be parameter-efficient while still capturing non-additive interactions. We highlight this distinction and clarify that our primary novelty lies in the compositionality formulation and triadic trend-labeled dataset, as well as an architecture tailored to this formulation under small-data constraints.
>
> ---
>
> ### **Ethics and consent**
>
> We acknowledge the ethical concern about using identifiable faces and voices. The raw videos come from the publicly available *The Weights Task Dataset* (Khebour et al., 2024). Syntality builds on this dataset by adding derived multimodal features and crowd-sourced annotations; we do not collect new recordings. We will make this provenance explicit in the Ethics and Dataset sections to alleviate any concerns.

---

> ### Author Response · Authors · 2025-11-22
> **Official Comment by Authors - 4**
>
> ### **Closing Remarks and Next Steps**
>
> Thank you again for taking the time to review the paper and providing helpful feedback! Do the above actions address your concerns with the paper? If not, what further clarification or modifications could we make to improve it?

---

### Official Review · Reviewer_AvSx · 2025-11-02

**Soundness:** 3
**Presentation:** 2
**Contribution:** 3
**Rating:** 6
**Confidence:** 3

**Summary:**

This paper introduces SyntalNet, a multimodal architecture designed to model emergent group-level (synchrony/confidence/interaction phase) and individual-level (leadership/engagement) social trends in small-group interactions. SyntalNet addresses the limitations of existing approaches, which fail capture inter-participant dependencies or handle modality and permutation challenges, through specialized modules for cross-person (SoSE-X), intra-modality (GSX), and cross-modal (GLRX) fusion.
To support evaluation, the authors construct Syntality, a benchmark dataset derived from the Weights Task Dataset, augmented with crowdsourced annotations for five constructs and formulated as a directional trend prediction problem (increase/stable/decrease).
Experiments show that SyntalNet significantly outperforms LSTM, Temporal CNN, and VLM baselines.

**Strengths:**

1.	Well-Motivated Architecture: The design of SoSE-X and GLRX is well motivated by the proposed need for permutation invariance and mask-aware multimodal fusion.
2.	Good Empirical Results: SyntalNet outperforms compared baselines across multiple constructs and metrics, with especially large gains for group-level constructs.
3.	Interdisciplinary Impact: The paper bridges social psychology, multimodal learning, and group cognition, an emerging and impactful direction for socially intelligent AI.
4.	Reproducibility: The authors commit to releasing code, features, and annotations, aligning well with the openness and reproducibility ethos.

**Weaknesses:**

1.Problem Formulation: Segmenting interactions into fixed 19-second clips for modeling group dynamic trends over one activity that lasts 9–34 minutes lacks validity. It is unclear whether such a short timescale meaningfully captures emergent group phenomena or introduces noise from overly limited temporal context.
2.Architectural Novelty: The proposed components appear to be adaptations of existing designs, and the paper lacks clear ablation studies demonstrating how each module contributes to performance or uniquely supports the modeling of group dynamics. The contribution appears engineering-heavy but conceptually shallow.
3.Evaluation Limitations:
a)The experimental baselines are relatively weak (logistic regression, LSTM, CNN). There are no comparisons against recent multimodal video understanding frameworks that can be far more relevant.
b)The Syntality dataset covers only a single collaborative task (block-weight inference), leaving the generalizability of SyntalNet to other forms of social interaction or team contexts uncertain.

**Questions:**

1.	Given that original sessions last up to 34 minutes, how do 19-second clips sufficiently capture higher-level constructs like group confidence or leadership, which often manifest at slower temporal scales?
2.	Can the authors provide ablation results or qualitative analyses to illustrate the impact of SoSE-X /GLRX and GSX? Also, how do these modules interact, are improvements additive or synergistic?
3.	While the appendix discusses statical associations with behavioral patterns, how do the proposed group trends relate to the existing behavior measurements on group collaboration, such as the Collaborative Problem Solving (CPS) Facets originally included in the Weights Task Dataset?
4.	Missing reference at line 146
5.	All figures in appendix are low-resolution.

---

> ### Author Response · Authors · 2025-11-22
> **Official Comment by Authors - 1**
>
> We thank Reviewer **AvSx** for the positive and encouraging assessment, particularly for recognizing the modular design, the strong group-level results, and the potential interdisciplinary impact of SyntalNet and the Syntality Dataset. We address the concerns about temporal formulation, architectural novelty, and evaluation scope below.

---

> ### Author Response · Authors · 2025-11-22
> **Official Comment by Authors - 2**
>
> ### **Problem Formulation**
>
> Our goal is to model *local trends* in individual and group constructs (confidence, synchrony, interaction phase, engagement, leadership) rather than to summarize an entire collaborative session with a single global label. Each 10 sec clip is a fixed-window "thin slice" in the sense of prior small-group interaction and cohesion work in the literature, where 8–21 sec segments have been shown to be sufficient for reliable judgments of cohesion and related group states (e.g., Panoptic-based cohesion studies using 8, 15, 21 sec windows). Building on this, annotators rate 10 sec clips on 3- or 5-point Likert scales; we then obtain trend labels by comparing the temporally smoothed, confidence-weighted posteriors of *pairs* of consecutive clips with a 1 s overlap, yielding an effective observation of 19 s. This timescale is long enough to capture short sequences of turns, mutual gaze, and micro coordination, while still being short enough that annotators can treat each clip as approximately homogeneous. Longer-term patterns emerge from the sequence of local trends rather than from a single long window. We will clarify this rationale and link it more explicitly to the thin-slice literature in the revision.
>
> ---
>
> ### **Architectural novelty and ablations**
>
> We thank the reviewer for raising the question of technical novelty. Our primary contribution is the **non-additive compositionality formulation** on group-state prediction, together with the **Syntality Dataset**, which makes this hypothesis empirically testable. Our experiments show that simply pooling individual-level signals leads to only modest gains over baseline models and performs substantially worse than approaches that explicitly capture cross-participant interactions. However, the common modeling assumptions and observed dynamics in the literature still heavily rely on additive formulations of group states. The non-additive compositionality formulation informs specific architectural requirements, which our other key contribution, the **SyntalNet** architecture, instantiates with three tightly coupled modules:
>
> The *Social Squeeze-and-Excitation-based Cross-participant Fusion (SoSE-X)* is not a standard SE block. Standard SE re-weights channels based on self-statistics. SoSE-X computes a permutation-equivariant "context" vector and conditions each participant's stream representation on this cross-participant signal via a residual gate. The learnable scalar residual coefficient allows the model to adaptively control cross-participant coupling strength. This yields an attention-like cross-participant fusion with far fewer parameters and lower data demands than self-attention.
>
> The *Branch-Separable Channel Mixer (GSX)* performs mask-aware fusion of substreams within a modality (e.g., facial landmarks, gaze, body pose, etc.). It handles intra-modality heterogeneity, allows exploiting complementary coverage of substreams, and remains robust to partial visibility and temporal misalignment.
>
> The *Gated Low-Rank Cross-modal Fusion (GLR-X)* extends low-rank bilinear fusion to multiple modalities by pairwise interactions between all active modalities, without the complexity of full cross-modal attention. It decouples unary vs. pairwise pathway with a learned mixing weight. It has an allocation gate that adaptively prioritizes modalties based on their quality/expressiveness.
>
> In the revision, we make this conceptual-to-architectural link more explicit in the Contributions section. We also add ablations that directly address the reviewer's concern. The ablations will include: (i) toggling SoSE-X and the group head to compare purely additive pooling, DeepSets-style pooling, SoSE-X–only, and the full model (SoSE-X + group head); (ii) stress testing GSX under temporal jitter, feature noise, and temporal band masking and compare with Concat+MLP and Mean Pooling; and (iii) comparing GLRX with Concat+MLP, Mean Pooling, a gated-sum-only, and a pairwise-only variant under modality dropout, noise, and shuffling. We also add an experiment to evaluate out-of-domain generalization in leave-one-group-out (LOGO) setting, where we report zero-shot performance on held-out teams and then measure how much performance recovers when finetuning the full model on small fractions of the target group or finetuning only the classifier heads with frozen encoder.
>
> Preliminary results show that (a) any removal or simplification of the compositional modules leads to systematic drops in group-level performance and robustness, and (b) under LOGO, SyntalNet exhibits a clear zero-shot performance gap due to covariate shift but recovers with modest adaptation, showing Out of Domain (OOD) transferability.
>
> ---

---

> ### Author Response · Authors · 2025-11-22
> **Official Comment by Authors - 3**
>
> ### **Baselines and Task**
>
> We agree that baseline quality is important. Logistic Regression is deliberately simple to act as the aggregation fallacy baseline, which treats the group as a bag of individuals without explicit relational modeling. The poor performance confirms our hypothesis that simple pooling is insufficient.
>
> However, our other baselines are not toy models. Mask-aware Temporal CNNs and BiLSTMs are widely used and competitive sequence models in human activity recognition and affective computing. In our setup, they share the same participant-indexed multimodal features input, roughly similar classifier structure, and training protocol as SyntalNet, follow a similar classifier structure, and are trained with the same optimizer, schedule, and regularization. We also tune their hidden size, depth, and dropout on validation folds. Therefore, they are roughly capacity-matched to SyntalNet. Conceptually, they represent strong temporal models that still rely on additive group composition.
>
> In addition, as a part of an ablation test, we include a DeepSets-style variant of SyntalNet, which aggregates participant embeddings with permutation-invariant pooling but removes explicit cross-participant coupling. Together with the VLM zero-shot comparison and the ablations, these baselines cover the main modeling assumptions we seek to test.
>
> Given the size of the dataset and the structured feature space, training large multimodal transformers would be impractical and highly prone to overfitting. We therefore believe this baseline suite is appropriate for the current dataset and task, and that the observed performance gaps reflect genuine benefits of SyntalNet's compositional design rather than overly weak baselines.
>
> Finally, regarding the generalization across social tasks, we fully agree that Syntality currently covers only one collaborative task. Our intent is not to claim universal generalization across all group settings, but to study non-additive compositionality in a controlled, physically grounded triadic problem-solving context where multimodal signals and task structure are well characterized. We now make this limitation explicit and report additional leave-one-group-out adaptation experiments showing that SyntalNet's representations can be adapted to unseen teams with modest amounts of target data, which we view as a first step toward broader generalization.
>
> ---

---

> ### Author Response · Authors · 2025-11-22
> **Official Comment by Authors - 4**
>
> ### **Response to Questions**
>
> > ***Given that original sessions last up to 34 minutes, how do 19-second clips sufficiently capture higher-level constructs like group confidence or leadership, which often manifest at slower temporal scales?***
>
>   Our labels reflect *local* trends rather than absolute levels. Prior work on cohesion and small-group perception shows that human raters can reliably infer constructs such as cohesion, engagement, and dominance from 8-21 sec thin slices. In Syntality, each trend label is derived from two partially overlapping 10 sec clips (effective 19 sec), and we sample such slices throughout the session. This allows us to capture short bursts of increased confidence or shifts in leadership (e.g., who proposes and defends solutions) while the sequence of trends over all windows reflects slower evolution.
>
> > ***Can the authors provide ablation results or qualitative analyses to illustrate the impact of SoSE-X /GLRX and GSX? Also, how do these modules interact, are improvements additive or synergistic?***
>
>   During the rebuttal period, we managed to conduct a number of various experiments. The new ablation on the contributions of components introducing non-additive compositionality show that: (i) removing both SoSE-X and the group classification head and predicting group labels by averaging individual logits (purely additive) yields the weakest group-level performance; (ii) adding either DeepSets-style pooling (group head only) or SoSE-X alone improves results, with SoSE-X giving larger gains; (iii) using both SoSE-X and the group head yields the best performance on group constructs. Similarly, replacing GSX and GLRX with simpler concatenation or mean-pooling baselines leads to systematic drops, particularly under temporal and modality corruptions. Overall, the improvements from SoSE-X, GSX, and GLRX are mostly additive, with the full model consistently strongest. We will provide the full details of these experiments in our revised manuscript.
>
> > ***While the appendix discusses statical associations with behavioral patterns, how do the proposed group trends relate to the existing behavior measurements on group collaboration, such as the Collaborative Problem Solving (CPS) Facets originally included in the Weights Task Dataset?***
>
>   The original Weights Task Dataset includes 19 CPS indicators aligned with three facets (constructing shared knowledge, negotiation/coordination, and maintaining team function). These indicators are sparse, event-like codes designed for educational and cognitive analysis, not dense time series suited as direct supervision for clip-level prediction. Our trend constructs (group synchrony, confidence, and interaction phase,  along with individual engagement and leadership) are complementary. Meaning that they capture continuous affective and interactional states that shape *how* CPS behaviors unfold.
>
> > ***Missing reference at line 146***
>
> We appreciate the pointers. We fix the missing reference at line 146.
>
> > ***All figures in appendix are low-resolution.***
>
> Thank you for bringing this up. We will replace all low-resolution appendix figures with higher-resolution versions.
>
> ---

---

> ### Author Response · Authors · 2025-11-22
> **Official Comment by Authors - 5**
>
> ### **Closing Remarks and Next Steps**
>
> We again thank you for your careful review and constructive suggestions. We hope the revisions and clarifications above directly address your concerns. Are there any remaining points you would like us to clarify or adjust further?

---

### Author Response · Authors · 2025-12-03
**Official Comment by Authors to All Reviewer and the AC**

We thank all reviewers for their constructive feedback. Below we summarize our main revisions and new experimental results.

**Novelty \& Positioning.** The field overwhelmingly uses additive composition of independently encoded individuals for group-state prediction. On Syntality, we show that such models systematically underperform architectures that encode cross-participant interactions, providing empirical evidence that additive composition is insufficient *in this triadic CPS setting*. We substantially expanded Related Work to include foundational studies by Gatica-Perez, Varni, Pantic, Vinciarelli, and Beyan et al. (2023).

**Testing the hypothesis.** Additive baselines (log-reg on pooled features, DeepSets head with SyntalNet Encoder) achieve average group-level macro-F1 ranging from 0.369 to 0.493, while that of baselines with cross-participant fusion in classification head only (TCN/BiLSTM) ranges from 0.466 to 0.537. However, models with explicit cross-participant fusion baked in architecture (SoSE-X + group head) achieve a macro-F1 of 0.623—a 26\% improvement over DeepSets additive baseline with the same Encoder. Main results use 5-fold random splits (macro-F1 = 0.622). Under leave-one-group-out, zero-shot macro-F1 drops to 0.378 on held-out teams, recovering to 0.471 when finetuning on 25\% of the target group; freezing the encoder and adapting only classifier heads reaches 0.582. In the revision we will report mean ± std across 5 folds for all methods and mark statistically significant improvements using paired t-tests across folds. SyntalNet instantiates minimal architectural requirements derived from this hypothesis: cross-participant fusion, mask-aware intra-modality fusion, and low-rank cross-modal interactions under small-data constraints.

**Temporal Formulation \& Label Semantics.** 10s clips follow thin-slice literature (8–21s); 3s stride; 19s observation. Labels are *directional trends* (increase/stable/decrease) from temporally smoothed posteriors—not absolute levels—enabling dense supervision of short-term dynamics.

**Dataset Scope \& Annotation.** We corrected positioning to acknowledge existing corpora. Syntality extends The Weights Task Dataset to provide triadic, participant-indexed streams with paired individual+group trend labels—a combination not jointly offered by prior datasets, enabling this specific hypothesis test. With more than 3000 samples and 1128 workers, Fleiss' $\kappa$ ranges from 0.28 to 0.47 for all labels and from 0.36 to 0.91 for strong labels, indicating fair to near-perfect agreement. We agree 10 triads limit external generalization; claims are restricted to this controlled setting.

**Baselines, Ablations \& Statistical Rigor.** We considered transformer-based video encoders (TimeSformer, VideoMAE) but their parameter counts made them prone to overfitting given limited number of clips from 10 triads. Our baselines (Temporal CNNs/BiLSTMs, DeepSets) test the additive vs. non-additive hypothesis at this dataset scale. Ablations confirm cross-participant fusion is critical (details in Testing section above).

**Clarity \& Artifacts.** Fixed broken references, uploaded sample videos. In the revision we will add confusion matrices and timeline plots, and improve Figure~1 to show data flow and operations explicitly.

---

> ### Author Response · Authors · 2025-12-03
> **Update: Ablation and Transfer Experiments**
>
> We have now integrated the ablation and transfer experiments into the **main manuscript**; below we briefly summarize what each experiment shows.
>
> **SoSE-X and group head (multi-person fusion).**
> We test four variants that either keep or remove cross-participant fusion in the encoder and/or in the prediction head. All purely additive baselines perform worst; adding only a DeepSets-style head yields a small gain, and introducing SoSE-X in the encoder gives the largest, consistent improvements across all constructs. This directly supports our claim that modeling inter-participant interactions is both necessary and more effective than post-hoc pooling (emergent team states cannot be reduced to pooled individual signals).
>
> **BSC-X** [previously GSX] **(multi-channel / intra-modality fusion).**
> We compare BSC-X against concatenation+MLP and mean pooling under temporal jitter, feature noise, and temporal band masking. BSC-X consistently attains the best scores and degrades slowest as corruption increases, indicating that its mask-aware fusion makes per-modality representations more robust when cues are misaligned or partially missing.
>
> **GLR-X (cross-modal fusion).**
> We freeze the rest of SyntalNet, swap GLRX with two alternative fusion blocks, and run a short corruption-augmented fine-tuning with modality dropout, noise, and cross-participant shuffling. Only GLR-X maintains strong performance under these stresses, showing that its low-rank cross-modal interactions capture complementary information between modalities more effectively than standard concatenation-based designs.
>
> **Out-of-distribution generalization (LOGO).**
> Finally, we show that while zero-shot transfer to a held-out team is understandably modest, even a small fraction of labeled clips from the target group yields substantial gains, and head-only fine-tuning with a frozen encoder plus more target labels closes most of the gap. This means that gaps stem from misaligned decision boundaries rather than missing social cues. Thus, it suggests that SyntalNet learns reusable group-dynamics representations rather than overfitting one set of triads, addressing reviewers’ concerns about generalization and transferability.

---

### Meta-Review · Area_Chair_6Tzz · 2025-12-03

**Summary:**

The authors propose a multimodal architecture to model emergent individual / group-level social trends in human interactions. The reviewers raised a great deal of concerns. Reoccurring topics across reviewers deal with missing related work, technical issues, small data set size, problem definition, evaluation protocol and so on.

**Reviewer Concerns:**

There are so many different issues found by the reviewers that it's difficult to tell whether they would be happy with the provided responses and revisions. Minor aspects and questions have certainly been clarified but I doubt that some other important aspects have been settled yet. This includes for example the small data set combined with the general problem definition or one of the reviewers beginning their review with "The paper seems to introduce...". In my view the paper needs a major revision to incorporate the reviewer feedback sufficiently.

**Reviewer Scores:**

The authors provided extensive responses to reviewer comments and questions. I truly believe that the authors did everything they could to change the minds of the reviewers by e.g., putting a lot of extra work in new experiments, scanning alternative data sets, related work,... and finally merging all this into detailed answers. However, there are so many issues with the submitted version that it's hard to believe that all this new information in the rebuttal would have led to a significant change in how the reviewers see the paper since it was pretty far from being matured enough for publication at ICLR at submission time.

---

### Decision · Program_Chairs · 2026-01-26

Reject